# Seasonal velocity patterns provide insights for the soft-bed subglacial hydrology continuum
Jane K. Hart [1] ✉, Nathaniel R. Baurley[1], Amy Bonnie[1], Benjamin A. Robson[2], Graeme Bragg[3] & Kirk Martinez[3]

Subglacial hydrology plays an important role in controlling glacier behaviour, influencing glacier retreat and the resulting contributions to sea level rise. Here we present a detailed seasonal data set from four soft-bedded temperate glaciers and demonstrate a continuum of subglacial hydrology from channelized to a multichannel distributed behaviour. Our results illustrate how this continuum may be affected by till grain size and subaqueous processes, and we quantify the relative timings of basal sliding and deformation. These different hydrologies have a distinctive seasonal velocity pattern, which although have been identified using a multi-data stream, we suggest can be classified using solely Sentinel-1 satellite-based glacier velocity data. The ability to categorize subglacial glacier hydrology over a much larger data set would allow a better parameterization of subglacial processes for ice sheet models.

As glaciers worldwide melt in response to climate change, it becomes increasingly vital to determine their present and future contribution to freshwater resources[1] and sea level rise[2]. These higher air temperatures will lead to increased melting of the glacier surface, generating additional meltwater which can flow directly to the glacier bed. There is, however, no linear relationship between melt and glacier response due to numerous factors[3] within the glacial system, but in particular the behaviour of water and till at the glacier base[4,5] which control basal sliding[6,7]. Glaciers terminating in a water body may be additionally affected by warmer oceans, increased calving and associated feedback processes[8].

Glaciers can flow over either hard (rocky) or soft (sedimentary) beds. It has been a longstanding assumption that the subglacial hydrology of rigid beds comprises a predominantly channelised drainage in the summer[9,10]. In contrast, it has been suggested that soft-bedded glaciers have a distributed system, based on observations from Antarctic ice streams[11–14] and mountain glaciers[15,16]. However, recent research on rigid-bedded glaciers from Greenland suggests that distributed drainage exists alongside channelised during the late summer[17], whilst simultaneously new examples of soft beds are being discovered across the ice sheet[18] leading to the hypothesis that many glaciers worldwide rest on mixed bedrock[7]. These observations highlight that the subglacial hydrology for all bed types is more complex than initially envisaged, and there is a need to identify different subglacial hydrologies to understand glacier behaviour.

Meltwater draining into the subglacial till changes the effective pressure (ice overburden pressure minus basal water pressure) of the glacier. As effective pressures range from low to high, so behaviour changes from sliding, to till deformation, to stick (no/little motion)[19,20]. It has been shown by numerous researchers that glaciers move via stick-slip motion[21,22], due to tidal or meltwater processes[23,24]. A study of Skálafellsjökull, a temperate soft-bedded glacier[16], showed that stick-slip motion occurred on a daily scale during the melt season and on a multi-day scale in winter. This process comprised both sliding and deformation, with the authors able to calculate the percentage times that these processes occurred throughout the season.

Understanding subglacial hydrology is important because numerous researchers have suggested that a distributed drainage has a higher water pressure and so lower effective pressure[9,10] than a channelised system, leading to reduced basal friction[25,26], resulting in faster surface velocity and increased glacier retreat. It has been suggested that this lack of understanding of subglacial process led to a total uncertainty of ~55% between different ice-sheet simulations associated with the Ice Sheet Model Intercomparison for CMIPS (ISMIP6)[27].

We examine a series of soft-bedded glaciers in maritime locations and use a range of techniques to investigate their seasonal behaviour. Our study and resulting predictions are applicable to glaciers with strong seasonality and summer melt. There is a continuum between multichannel distributed and channelized systems and the dominant controls on the continuum are

[1]School of Geography and Environmental Science, University of Southampton, Southampton, SO17 1BJ, UK. [2]Department of Earth Science, University of Bergen and Bjerknes Centre for Climate Change, 5020 Bergen, Norway. [3]Electronics and Computer Science, University of Southampton, Southampton, SO17 1BJ, UK. ✉e-mail: j.k.hart@soton.ac.uk

till grainsize and subaqueous processes. The distributed system is characterised by dramatic speed-up events associated with warm days during winter (winter events), a delayed and slow spring event, relatively stable summer velocities, and peak velocities during autumn. This summer pattern occurs because an anastomosing system develops which can adapt to meltwater changes, promoting stable velocities and facilitating water storage. The water is released during the winter allowing winter speed-up events, with a period of recharge needed during spring. In contrast, the channelized system has few/no winter speed-up events, a fast-spring event at the beginning of spring, a variable velocity over the summer (with a peak at the beginning or middle), and similar high velocities during autumn. Within a channelized system, the connected area increases over the summer (reducing velocity), but has much lower water storage, hence fewer winter events, and a dramatic spring event once air temperatures and meltwater increase. We are also able to quantify subglacial processes and use these, alongside the hydrological findings, to predict till and glaciofluvial sedimentology associated with different soft-bedded hydrologies. In addition, using the results from our multi-data stream analysis, we propose that it is possible to identify the subglacial hydrological systems from the analysis of Sentenal-1 data alone.

## Results
### Field sites

We report results from four temperate soft-bedded glaciers in relatively mild maritime locations (Fig. 1), using field based and remotely sensed data. We have studied three adjacent glaciers from the Vatnajökull ice cap, Iceland: Skálafellsjökull[16,28], Fjallsjökull and Breiðamerkurjökull. All three calve into proglacial lakes, approximately 20 m, 100 m, and 300 m deep respectively[29,30]. The fourth glacier is Briksdalsbreen, Norway; which is an outlet glacier of the Jostedalsbreen ice cap[31], and also previously calved into a 20 m deep proglacial lake. The field study of Briksdalsbreen took place during 2004-6, and since then the glacier has dramatically retreated onto a

plateau, so we have derived the remotely sensed velocity from the nearby Nigardsbreen (more information below).

Details about the sites are shown in Table 1. All four glaciers rest on a till base and have a foreland geomorphology composed of flutes and push moraines, indicative of subglacial deformation[32]. Field data was collected via the Glacsweb environmental sensor network[33] which comprises sensor nodes that relay their data to a sensor network server in the UK. This network included in situ wireless sensor probes inserted into the till (0.16 m long) which contained micro-sensors measuring water pressure, probe deformation, resistance, tilt and probe temperature[16,34]. We use the water pressure results as part of the effective pressure calculations (ice overburden pressure minus basal water pressure), measured in hydraulic head (m) and then compared these with the water column height from the known glacier thickness (flotation pressure %). The observation system also included geophones to measure ice-quakes[35]. Secondly, we designed and built a web connected real time kinematic GNSS which was able to send a 'live' data stream back to a server[36]. Details of sensors, readings, locations and errors have been discussed previously[16,28], and the data sets are outlined in the Method section.

Figure 2 shows an annual pattern of air temperature, effective pressure, and daily surface velocity for each glacier (where available). The data presented represents the year with the fullest data set for each glacier respectively. The data is plotted from 1ˢᵗ September (DOY 244) 2009/10 for Skálafellsjökull, 2017/8 for Fjallsjökull and Breiðamerkurjökull (with July and August velocity data from 2023 for the latter due to data availability), and 2004/5 for Briksdalsbreen. At Briksdalsbreen the surface velocity was only available monthly.

The annual 12-day velocity data from Sentinel-1 SAR imagery for each glacier for 2018/19 is illustrated in Fig. 3. One year was chosen for comparative purposes. The variation each year, for each glacier was low (standard deviation of as percentage of the mean; Skálafellsjökull 9.76%; Fjallsjökull 14.64%, Breiðamerkurjökull 7.02%, Briksdalsbreen 7.55%), so the pattern for each year is similar.

A difference is expected between the GNSS and remote sensing velocities due to the difference in temporal and spatial coverage. The GNSS results reflect daily measurements from a point source whilst the remote sensing velocities are averaged over 12 days, and across the centrelines of each glacier (see Methods for more detail). We compare the pattern of surface velocity change from the two techniques rather than the absolute values.

We have defined the seasons for the four glaciers based on air temperature, which is related to melt[37]. Since these glaciers are in different locations/altitudes the detailed thresholds are discussed in the Methods.

### Skálafellsjökull

The annual pattern of effective pressure and velocities are shown in Figs. 2a and 3a, and details of till water pressure, tilt and velocity for a multi-day event in winter and diurnal change in summer are shown in Fig. 4a and b. Winter comprises two states. A base state building to a high till water pressure and low velocity, and speed-up events where water pressures dramatically decrease, and velocities increase. These speed-up events also occur every time the temperature rises above zero, 83% of the time for the GNSS data and 84% of the time for the 12-day velocity data (see Methodology for details).

During these events, it has been reported that there is a vertical rise of the glacier and high meltwater discharge[28]. At the same time there is a pattern of tilt change in the probe comprising an initial antithetic (backwards) tilt change and then synthetic (forwards) tilt change[16]. During the spring there is also a velocity increase, but this is within the upper range of the winter event velocities. Each year the spring speed-up occurs ~17 days after the beginning of spring (see Methods for derivation of the seasons) and is not related to a specific weather event. In summer, both water pressures and velocity remain consistently high (summer remotely sensed velocity variation s.d % mean=5%). In contrast, till water pressure slowly decreases during autumn, but velocity peaks are some of the highest. The effective pressures are low in the summer and high in the winter and have a very similar in pattern at each probe.

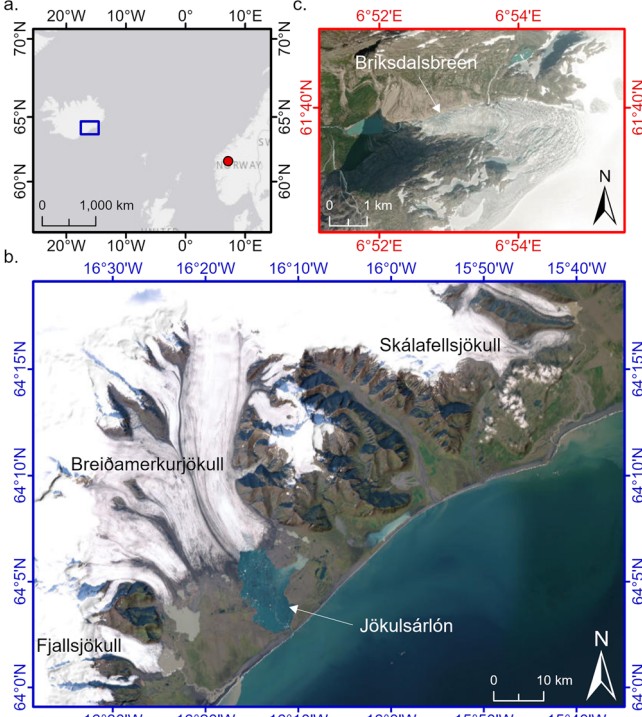

**Fig. 1 | Details of the field sites. a** Overall location. **b** Fjallsjökull, Breiðamerkurjökull and Skálafellsjökull, Iceland. **c** Briksdalsbreen, Norway (Source: Esri, Maxar, GeoEye, Earthstar, Geographics, CNES/Airbus DA, USDA, AreoGRID, IGN and the GIS User Community).

**Table 1 | Details about the four sites**

| Field Site | Mean annual air temp. (°C)[a] | Glacier area (km²) | Altitude of study site (m) | GPR | In situ Probe data | Velocity | | Till mean Grain size (Wentworth Class) |
|---|---|---|---|---|---|---|---|---|
| | | | | | | GNSS | Sentinel -1 | |
| Skálafellsjökull 64.251°N, 15.832°W | -0.1 | ~100 | 792 | ✓ 2008 2014 | ✓ 2008-2013 | TOPCON legacy 2008–2012 Leica 1200 2012/13 | ✓ 2017–2022 | 53 µm Silt to very fine sand |
| Fjallsjökull 64.011°N, 16.428°W | 5.1 | ~45 | 89 | x | x | Smart tracker 2017–2020, 2023/4 | ✓ 2017–2022 | 178 µm Fine sand |
| Breiðamerkurjökull 64.099°N, 16.324°W | 4.6 | ~900 | 173 | ✓ | x | Smart tracker 2017–2022 2023/4 | ✓ 2017–2022 | 474 µm Silt to very fine sand (overlying up to 50 m gravels) |
| Briksdalsbreen 61.663°N, 6.864°E | 5.7 | ~15 | 375 | ✓ | ✓ 2004-2006 | TOPCON legacy 2004–2006 | x | 1600 µm Very coarse sand |
| (Nigardsbreen 61.884°N, 7.193°E) | (1.9) | (~48) | | | | | ✓ 2017–2020 | |

[a]Data from the four sites over the study period, taken from the local metrological station and corrected for altitude/local conditions using the weather station on the respective base station. Glacier details: Skálafellsjökull[28], Fjallsjökull[28], Breiðamerkurjökull[4,69], Briksdalsbreen[31], Nigardsbreen[70].

Figure 4b illustrates the mean melt season diurnal tilt change, air temperature and velocity patterns. Air temperatures were lowest at 07:00, rising to a high plateau 12:00-17:00, cooling rapidly until 21:00, then cooling more slowly until 07:00, with a small rise at 04:00. Velocities were also low in the morning, rising to a peak at 13:00, and then decreasing throughout the afternoon and evening, before a speed-up event at 4:00. The tilt-changes are generally low during the main high velocity event at midday, but high during the slow down and speed-up period. These high tilt changes correlate with a large number of subglacial ice quake events[16].

A GPR survey of the site revealed a multichannel system[38] and that during summer discharge only accounted for approximately 60% of inputs, whilst in winter discharge was five times any inputs. As a result, we suggest that Skálafellsjökull has a distributed subglacial hydrology. Excess meltwater enters the aquifer, subglacial till, and the distributed system itself in a series of backwater reservoirs. Water from the till and the reservoirs is then partly released during winter during a series of speed-up events.

### Fjallsjökull
The annual velocity patterns are shown in Figs. 2b and 3b. During winter there is a distinct base velocity, with speed-up events ( ~ six times faster than the base), predominantly related to high air temperatures (temperatures >5 °C), for both the daily (77% correlation) and 12-day (80% correlation) velocity. There are an average of 11 events per year (based on the GNSS data), with an average length of 17 days (based on the GNSS data). Detail of one of these events is illustrated in Fig. 4c.

There were speed-up events ~13 days after the beginning of spring, but these are similar to the smaller winter event speed-ups which mark the beginning of the melt season when velocities rise to a higher mean level. Summer velocities are relatively stable (summer remotely sensed velocity variation s.d % mean = 3%). Velocities were also high during autumn, likely to be related to weather conditions.

The mean summer diurnal pattern of velocity and air temperature is shown in Fig. 4d (mean of 2023 results). Air temperatures are lowest at 04:00, slowly rising throughout the morning to a high temperature plateau between 13:00–15:00 and then decreasing overnight. Velocities are also lowest in the early morning (06:00), rising to a peak at 15:00, and then decreasing overnight. The peak velocity occurs ~1-2 h after peak air temperatures. Note the air temperature data is hourly whilst the velocity data is every 3 h.

### Breiðamerkurjökull
At Breiðamerkurjökull there are fewer and relatively slower winter speed-up events (air temperatures >5 °C) (approx. three times faster than the base

velocity) (Figs. 2b & 3b), and many are apparently unrelated to meteorological conditions. The correlation for the daily velocity and air temperature is only 61% and 58% for the daily and 12-day scale, respectively. When a speed-up event does occur in response to increased air temperatures, it occurs almost immediately (Fig. 4c) and there are often other speed-up events within one winter event.

There was a speed-up event at the beginning of spring associated with the rise in air temperatures, marking the beginning of the higher melt season velocities. The velocity increase was recorded in the daily velocity from 2018, and occurred on the 8th May (DOY 128), which was equivalent to the 90% percentile of the winter events, followed by a much faster speed-up event on the 13th May (DOY 133) (140% larger than fastest winter event). This latter event occurred five days after the beginning of spring. In the 12-day data this speed-up occurred once air temperatures rose above the spring threshold, with velocities either equal to maximum winter event velocities (in 3/5 years) or greater than the winter event velocities (2/5 years). Summer velocities, meanwhile, are relatively stable (summer remotely sensed velocity variation s.d % mean = 1%), whilst autumn velocity patterns are intermediate between summer and winter.

At Breiðamerkurjökull, the diurnal velocity pattern (mean summer 2023) is different to that observed at Fjallsjökull (Fig. 4d). The velocity has a double peak, rising during the afternoon to a peak at 15:00 (2 h after peak temperature), then decreasing during the evening before rising to a second peak during the night (03:00).

### Briksdalsbreen
Figure 2c shows the annual air temperature (2004/5) plotted against the effective pressure from two probes: B8 (2004/5) and B12 (2005/6), as well as monthly velocity data. These two probes were inserted into the till in different years and slightly different locations[31]. The in-situ tilt data from these probes[31] was used to investigate subglacial processes. The details of till water pressure and tilt for a multi-day event in winter and diurnal change in summer are shown in Fig. 4e, f. Previous GPR surveys of the site[31] indicate that Briksdalsbreen has a channelized subglacial hydrology.

The field study at Briksdalsbreen took place before Sentinel-1 was launched, so there are no available satellite images of the glacier from 2004/5 with which to calculate velocity at a comparable temporal resolution. Additionally, given the extensive retreat of Briksdalsbreen we opted to use utilise Sentinel-1 based velocities of the nearby Nigardsbreen glacier, as it is of a similar size and altitudinal range to Briksdalsbreen during the early 2000's. We also assume Nigardsbreen has a similar channelised hydrology as it has a large portal at the margin, which is similar in appearance to that observed at Briksdalsbreen during the field study.

**Fig. 2 | Annual records. a** Annual pattern of behaviour (effective pressure, daily velocity, air temperature): a) Skálafellsjökull, Probe S21 and S25, 2009/10. **b** Fjallsjökull (upper) and Breiðamerkurjökull (lower), daily velocity 2017/18. **c** Briksdalsbreen, Probe B8 (solid line) and air temperature 2004/5 and B12 (dotted line) from 2005/6, monthly velocity data (mean 2003-06) (all data plotted from 1st September).

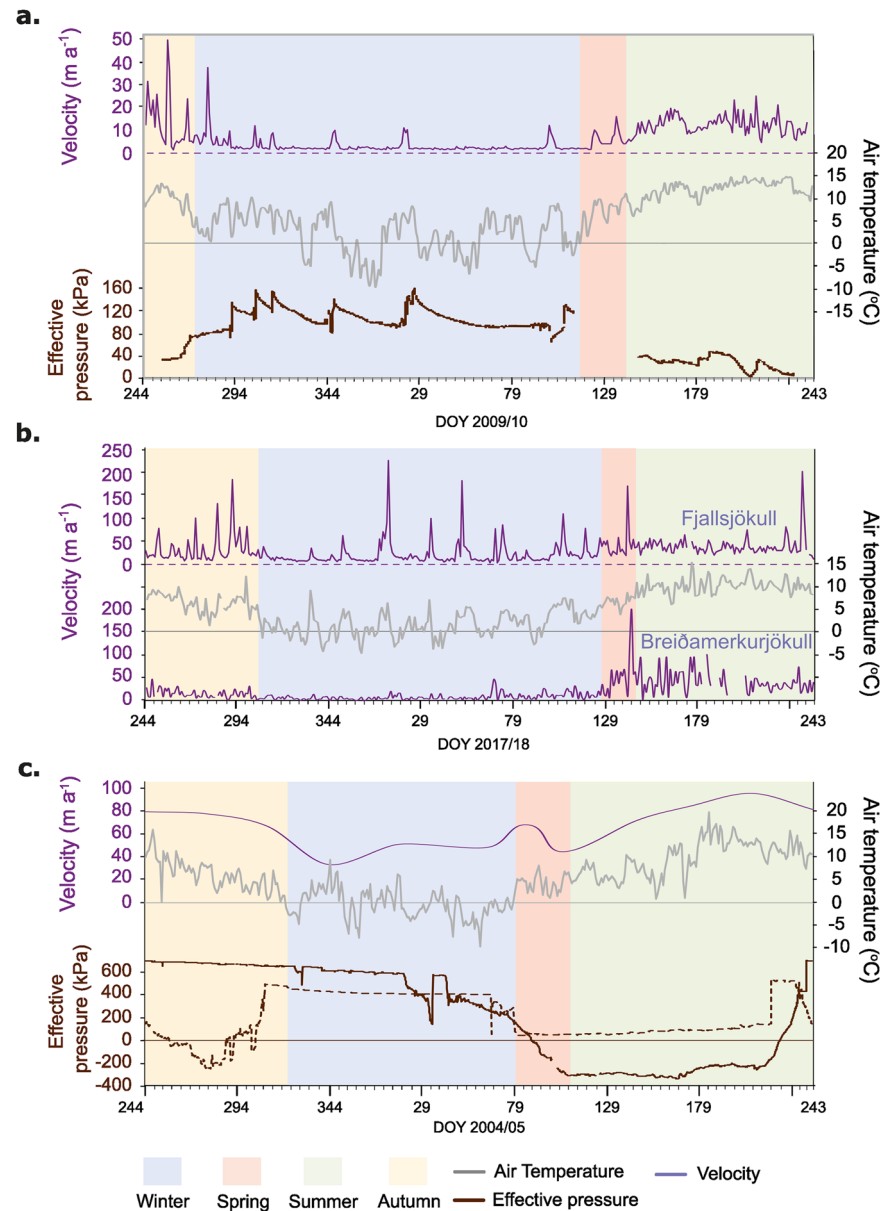

During the winter, the effect pressure is high for probe B12, whilst at B8 it is generally high, except for a series of dramatic increases that occur when air temperature declines below zero. There were four events during 2004/5: 16th November to 17th December (DOY 321 -352), 18th December to 18th January (DOY 353-18), 19th January to 11th February (DOY 19-42), and 12th February to 21st March (DOY 43–80), with details of one of these events illustrated in Fig. 4e. Initially, there is a period where air temperatures are below zero, followed by a phase of above-zero air temperatures. Once air temperatures fall below zero, there is a mean four-day lag before water pressures increase. This behaviour is accompanied by an antithetic (backwards) tilt movement lasting one day and then a large synthetic (forward) tilt movement, also lasting one day, followed by a period (mean length 16 days) of low synthetic tilt during sub-zero air temperatures. Once air temperatures rise above zero, water pressures rapidly declined, and there is a repeat pattern of one day antithetic tilt followed by one day of synthetic tilt. After this, there is a long period (mean length 18 days) when the water pressures are low, with low synthetic tilt.

During the spring, effective pressures decrease as air temperatures increase. For probe B8 the decrease is relatively slow (14.4 kPa decrease per day), whilst at B12 there is a double event. A one-day dramatic

decrease and increase on DOY 68, followed by an abrupt decrease on DOY 86 (98.3 kPa per day) which marks the spring event. A similar pattern is also seen at probe B10 (not shown here but reported elsewhere[31]), which shows a double peak but with different timings (initial decrease and rise DOY 106, main abrupt decrease DOY 117). The spring event is also observed in the monthly GNSS record. Summer begins with low effective pressures ( ~ DOY 111– 175), which slowly increase in mid-summer ( ~ DOY 176–223), then rapidly increase in late summer ( ~ DOY 224–240). At probe B8 during summer, the water pressure exceeds the local overburden pressure (negative effective pressure), known as excess water pressure, whilst at probe B12 the effective pressure is low (but positive). Summer velocities are high, with peaks in June and July. Similarly, this velocity pattern was also observed in measured stake data from 1996–2000[39]. During the autumn (DOY 241-320), effective pressures are either high (B8) or low (over pressurized) (B12).

Figure 4f shows the mean summer air temperature and tilt. Air temperatures rise in the morning (04:00 to 12:00), peak in the afternoon (12:00 to 16:00) and then decline overnight (16:00-04:00). The tilt change pattern shows least movement in the night (04:00), slowly rising in the morning (04:00–12:00), peaking at 16:00, and then falling. The autumn tilt changes

**Fig. 3 | Annual pattern of 12-day velocity derived from Sentinel-1 plotted against daily air temperature 2018/19. a** Skálafellsjökull. **b** Fjallsjökull (lower graph) & Breiðamerkurjökull (upper graph). **c** Briksdalsbreen (velocity data from Nigardsbreen) (all data plotted from 1st September; horizontal bar indicates 0°C).

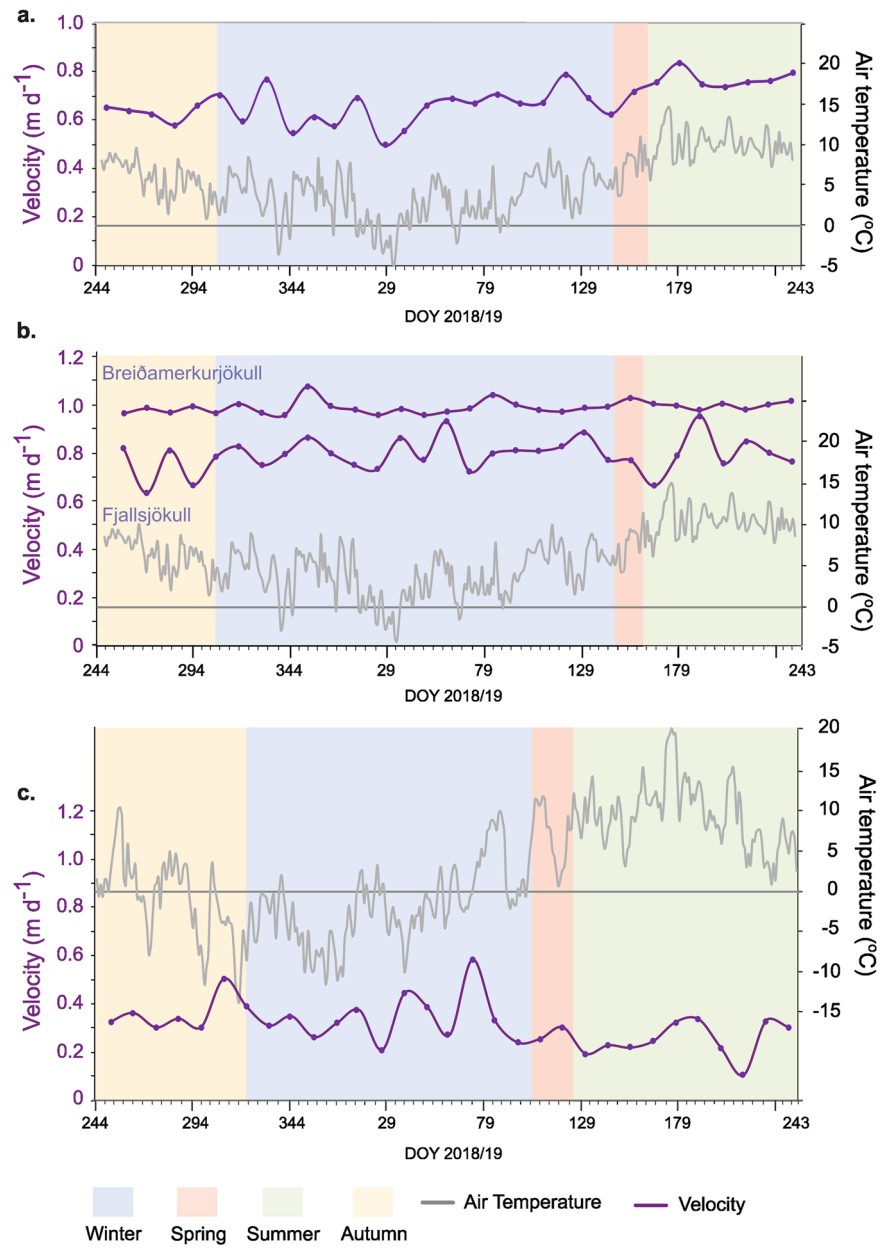

from probe B12 show a similar pattern, but with more extreme changes, with a secondary peak in tilt during the early morning (04:00).

In winter, the 12-day velocity data (Fig. 3c) vary throughout the season, but with a low (~ 54%) correlation with temperature data. In spring, there is a fast speed-up event equal to or greater than the winter events, which occur during the first 12-day period of spring each year. Velocities tend to be variable over the summer with either a peak at the beginning followed by general decrease or a peak in the middle of summer (summer remotely sensed velocity variation s.d % mean =28%), before rising again in autumn.

## Discussion

The growing recognition of multichannel distributed drainage systems associated with soft-bedded glaciers[7,11–14] could lead to a possible assumption that all soft-bedded glaciers have such a system. However, we have demonstrated here that this is not the case, and that there is likely a continuum between a distributed and a channelized system (Fig. 5). We will now discuss the different seasonal behaviours associated with this continuum, suggest some controlling factors, outline how it is possible to

identify the separate regimes based on Sentinel-1 velocity records, and quantify and identify different subglacial sedimentary processes.

The results from Skálafellsjökull provide evidence of the seasonal development of a distributed river system associated with a soft-bedded glacier (Fig. 5a). During the summer, the level of anastomosing was related to melt, and large parts of the bed had high connectivity. At the beginning of summer, as the channels were opening, increases in melt led to subsequent increases in velocity. However, later in summer, the high levels of melt were able to be accommodated by the hydrological system, resulting in an enhanced anastomosing velocity decline. Whenever the melt exceeded the carrying capacity of the system, however, this led to reduced effective pressure and summer speed-up events[40,41].

During autumn meltwater inputs decreased, meaning the level of anastomosing (and, therefore, the connectivity) also decreased, with water flow concentrated along the main channels. As a result, water may have become isolated in backwater elements ('ponds'), whilst it could also drain out of the till resulting in decreased water pressures. High meltwater inputs at this time resulted in the fastest peak velocities because in this transitional state the subglacial system was easily overwhelmed[40,41].

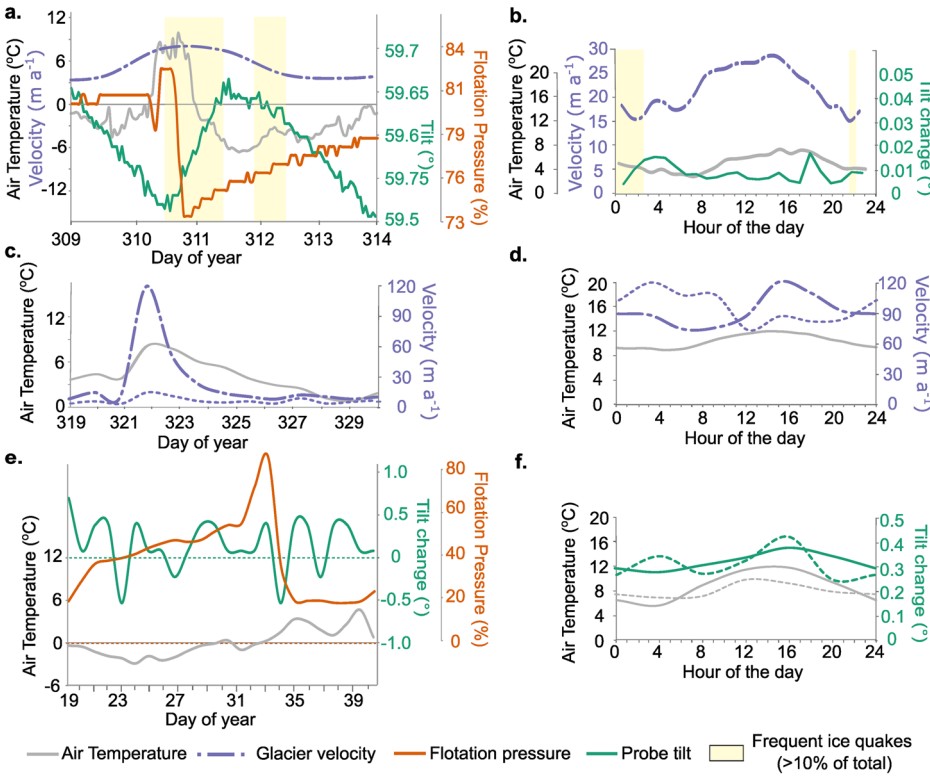

**Fig. 4 | Examples of winter events (left side) (note change in daily scale) and mean diurnal summer properties (right side). a** Skálafellsjökull (DOY 309-314, 2012) (floatation pressure estimated from the behaviour during winter events during 2009). **b** Skálafellsjökull summer 2012. **c** Fjallsjökull (dashed line) and Breiðamerkurjökull (dotted line) (DOY 319-330, 2018). **d** Fjallsjökull (dashed line) and Breiðamerkurjökull (dotted line) summer 2023. **e** Briksdalsbreen (DOY 19-42, 2004/5). **f** Briksdalsbreen summer (probe B8) (DOY 211-248, 2004/5) (solid line), and mean autumn (DOY 249-320, 2005/6) (dotted line).

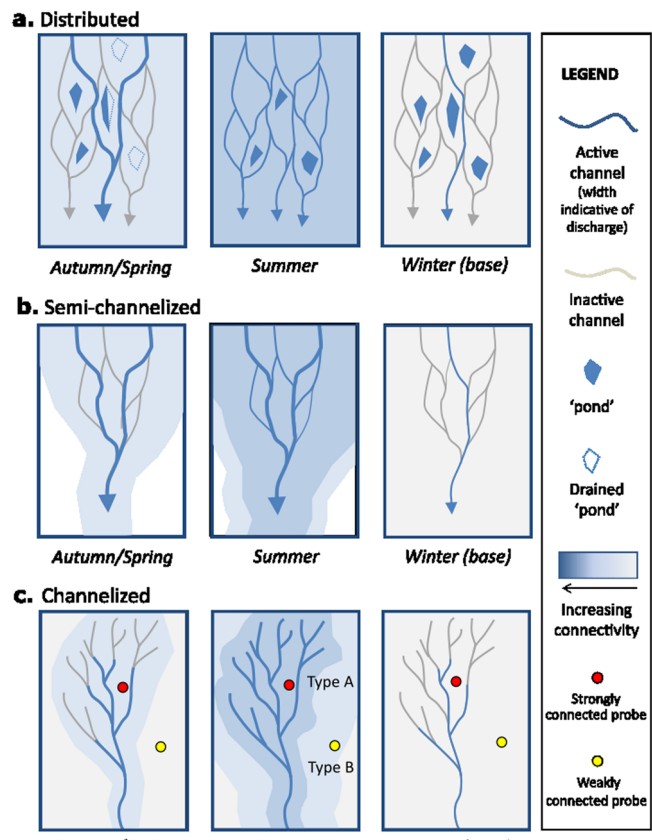

**Fig. 5 | Conceptual diagram to illustrate the seasonal subglacial drainage associated with soft-bedded glaciers. a** distributed system. **b** demi-channelized. **c** channelized system. The 'ponds' are representative of the larger subglacial reservoirs.

During winter, as there was little meltwater generated, there was low flow through the main channels and the till, resulting in low velocities. However, during winter events there were speed-up events with shear-induced till dilation[20], glacier uplift and high discharge. Water was released from the subglacial reservoirs, which included the till, cavities, macroporous sources and ponds. These became 'connected', and water was able to flow at the ice/till interface into the main channels.

During spring the meltwater input increased, being accommodated within the main winter channels whilst connectivity within the till increased, until a threshold was passed (after ~17 days), and then a speed-up (spring) event occurred. Afterwards, the system adapted to the new higher melt levels associated with summer via the anatomising of the channels.

The pattern of behaviour at Fjallsjökull is very similar to that at Skálafellsjökull, with the winter speed-up events, a non-weather-related spring event occurring ~13 days after the beginning of spring, relatively stable summer velocities, and high autumn velocities. Because of this similar velocity pattern, we suggest that this glacier also has a distributed subglacial system.

Breiðamerkurjökull is adjacent to Fjallsjökull with similar weather and bedrock conditions and thus would be expected to have comparable glacier behaviour. However, there are several key differences. This includes a lack of winter events, a distinct spring event related to weather conditions, and a dual peak in summer diurnal air temperatures (12 h apart), whilst there are also numerous small speed-up events unrelated to weather conditions that occur during both summer and winter. We suggest that the difference in response is due to a combination of the presence of the deep proglacial lagoon Jökulsárlón as well as high meltwater flux and subglacial transmissivity at Breiðamerkurjökull. It has been shown that deep lakes generate high hydrostatic pressure which leads to lake water being pushed up-glacier into subglacial channels[42,43]. At the same time, high summer air temperatures and rainfall will generate high discharge. Towards the margin, the bed could become over pressurised to enable the meltwater to be evacuated from the glacier. It has also been suggested that there some perennial large subglacial channels due to high meltwater inputs from the large scale of the glacier and the transmissibility of the glacial fluvial sands underlying the till[44].

**Table 2 | Criterion to identify subglacial hydrology based on seasonal velocity changes**

|  | **Distributed** | **Semi-channelized** | **Channelized** |
|---|---|---|---|
| *Winter* | Winter events | No/few winter events | No/few winter events |
| *Spring* | 'Slow' speed-up event ~15 days after the beginning of spring | 'Fast' speed-up event at the beginning of spring | 'Fast' speed-up event at the beginning of spring |
| *Summer* | Stable velocities | Stable velocities (+/- 5% from mean) | Variable over the summer ( > 5% variation from mean)) |
| *Autumn* | High velocities | | |

As a result of a combination of these processes, we suggest that the distributed system switches to a more effective channelised system when meltwater input is high, releasing the excess melt water in short bursts. This allows any summer excess meltwater to be drained into the lake rather than be stored with the subglacial system. In addition, calving associated with the interactions between the glacier margin and the proglacial lake may help explain the non-weather-related speed-up events[45]. Increased velocity would encourage crevassing, allowing additional meltwater to reach the bed, as well as increasing the buoyancy of the glacier margin.

Although Jökulsárlón has a tidal influence with two high tides a day[46], which could potentially affect the glacier, these occur on average 12.5 h apart ( + 40 mins/-31 mins), moving on a mean 48 min ( + 25 mins/-14 mins) per tide cycle, so even out over the season. As such, we suggest the dual peaks recorded in the summer diurnal pattern likely reflect the dominance of two water pathways. There is an initial velocity increase associated with midday melting, but much of the water is prevented from draining due to high hydrostatic pressures from the lake. This excess water builds up during the night until a threshold is passed, after which channelized drainage dominates.

We suggest the following seasonal pattern (Fig. 5b): During summer there is a multichannel river system, but when melt is high, channelization occurs, which also drains any storage. In autumn, the anatomising decreases to a few main channels. During winter, these channels continue to shrink, but due to the lack of storage during times of melt, there is limited glacier response in terms of speed-up events. During spring the onset of high melt associated with warming causes a dramatic spring event, which cause the subglacial system to be overwhelmed[40].

In contrast, Briksdalsbreen reflects typical channelized behaviour (Fig. 5c), with the effective pressure data indicating two distinct regimes within the system. The results from probe B8 reflect 'highly connected' behaviour from a site close to the main channel (Type A) whilst those from B12 reflect 'weakly connected' behaviour away from the main channel (Type B).

At the beginning of summer (for both Type A and B behaviour), the effective pressure is low, however, as the summer progresses effective pressures slowly increase. There are high velocities at the beginning of summer, followed by a velocity decline as the summer progresses. This is similar to the pattern described from Greenland, where in early summer, meltwater is able to lubricate the bed, resulting in basal sliding[47], whilst later in summer, this meltwater can be accommodated by the subglacial system via the growth of the 'weakly connected' distributed drainage[10,17] so velocities are reduced[48,49].

As autumn approaches, close to the main channels, the effective pressure rises as these channels reduce in size in response to reduced meltwater inputs. However further away from these channels, the increase in effective pressure is slower as the system takes longer to adapt. Once autumn is established, any warm day (with air temperatures above the stated thresholds) will result in dramatic increases in velocity as the system is easily overwhelmed by meltwater inputs. Close to the main channels these high meltwater inputs drain into the channels and connected till, but away from the main channels these water inputs cannot easily drain, and so effective pressures decreases.

During winter, air temperatures remain above zero, so a small amount of water is generated by melt, as well as heat from glacier movement, and so

the conduits remain open, although with much lower discharge. This results in low till water pressures, although some meltwater can drain through the till into the main channels. However, when air temperatures are sub-zero, melt decreases, and the conduits begin to close. This means sites close to the main channel have their drainage restricted and water pressures rise, whilst those further away do not change, as any change in melt is distributed evenly through the till.

During spring, the area surrounding the main channel slowly adjusts to the increase in water input associated with spring melt (Type A), as parts of the main channel may not have completely closed during winter. In contrast, sites further from the main channel have a more dramatic response to meltwater input, as the area will become rapidly connected (Type B). This will occur at different times, associated with water accumulation and specific water passage through the till. At first, the channels cannot accommodate the extra discharge, and so there is a rapid decrease in effective pressure, followed by an increase in effective pressure. This could be associated with shear induced till dilation and possibly a speed-up event[20]. As the meltwater continues to increase with the onset of spring effective pressures decrease again again and the system becomes adapted to the high melt and effective pressures can remain low.

There have been numerous theoretical attempts to establish the characteristics that determine whether a channelized or distributed system will form at soft beds[50–52]. These previous attempts have investigated the relative properties (and importance) of ice, sediment, and ice surface slope. However, we suggest that our results indicate the importance of sediment grain size. The subglacial till at Skálafellsjökull and Fjallsjökull is relatively fine grained (ranging from coarse silt to fine sand) and has a low porosity and transmissivity, which results in the development of the multichannel form and a spatially similar low summer effective pressure pattern. In contrast, at Briksdalsbreen the presence of coarse-grained till (very coarse sand) indicates that at the beginning of the melt season, as the distributed system grows, a low-pressure channel is formed from sediment erosion. This till has a relatively high porosity and transport capacity and so rapidly draws water from the surrounding till, allowing this channel to remain relatively stable[52]. This results in spatial variation in subglacial till water pressures across the bed, including areas of over pressurization (as described above). A similar situation is found at Breiðamerkurjökull, where the early summer distributed system probably remains up-glacier, but towards the margin develops into a channelised system due to a combination of the coarse till, high discharge and deep proglacial lake.

We propose that the members of the subglacial hydrology continuum have a set of distinct velocity patterns that may be used to identify the subglacial hydrology in regions where air temperatures periodically rise above zero during winter (Table 2). Speed-up events in winter and/or spring that were identified in the GNSS record can also be seen in the 12-day Sentinel-1 record. The difference between the distributed system and the better-known channelized system is as follows: Since water is stored in the multichannel system itself, this is released in a series of winter events which are either absent from the channelised system, or fewer in number. By spring, much of the stored water in the distributed system has already been released, and so a period of spring melting is required to generate enough water for a speed-up event. In contrast, in the channelized system the spring inputs immediately overwhelms the capacity of the system, resulting in a fast and distinct speed-up event. In summer, in the distributed system, the

constantly changing anastomosing allows the velocity to remain relatively stable, whilst in the channelized system the development of the weakly connected drainage reduces the velocity (resulting in an overall variable velocity). However, by the end of summer both systems are relatively similar so autumn velocity increases are observed in both.

Since Breiðamerkurjökull has a semi-channelised subglacial hydrology, it demonstrates a mix of the criterion. As it has low storage, there is limited response to winter events and a dramatic early spring event similar to the channelized system. However, during summer it behaves in a similar way to the multichannel distributed system and has a relatively constant summer velocity as a result.

Meltwater-driven stick-slip motion was observed at Skálafellsjökull throughout the year and comprised of four phases. During the melt season this motion occurred on a diurnal scale, with meltwater entering the system in the morning, which continually increases until a threshold is crossed, at which point the subglacial hydrology is overwhelmed, and there is a period of glacier sliding (Phase 1) (high velocity and little change in tilt). Subsequently, the glacier reconnects with the bed and there is deformation (Phase 2) (high tilt change). As air temperatures and meltwater decline in the afternoon/evening, velocities are at their slowest and deformation is low (Phase 3), and then as air temperatures begin to warm the next morning, the glacier begins to speed up and deformation increases (Phase 4) (high tilt change). In winter, there is a similar multi-day pattern associated with meltwater from the winter events. The sliding phase is accompanied by a decline in water pressure and antithetic (backwards) change in tilt associated with unloading[53,54] (Phase A). The reconnection phase has very high synthetic tilt (Phase B), and then there is a period of stick with no tilt movement as water pressures fall below that able to produce deformation (Phase C). Finally, there is a final phase of increasing water pressures and synthetic tilt movement (Phase D).

We extended this interpretation to the other glaciers in the study (see Methods) and based on these observations it was possible to calculate the amount of time that these processes occur in both summer and winter, and over the whole year. We divided the time into three main states: i) sliding (Phase 1 & A); ii) deformation (Phases 2-4 & B, D); and iii) no deformation (Phase C). In this way we have a quantification of subglacial processes (Table 3).

The pattern at Fjallsjökull was very similar to Skálafellsjökull. However, at Breiðamerkurjökull, during summer there is a double diurnal velocity peak and in winter 81% of the winter events were accompanied by more than one sliding event, many of which were not related to weather conditions, whilst 55% of cycles did not have a sliding phase at the beginning. At Briksdalsbreen the pattern in the melt season was similar to Skálafellsjökull, but in winter there were two speed-up events during each winter event. The first is associated with the build-up of porewater pressure due to the restriction of water moving through the till. The second is due to melt water production associated with rising air temperatures. In this way Briksdalsbreen has two sliding episodes in each winter event, but the number of events per year is very low, so there is only 6% sliding during the winter.

Overall, our study glaciers have a relatively similar annual pattern, which suggests that deforming bed glaciers, irrespective of their subglacial hydrology, have a typical pattern of sliding occurring ~15% time, deformation ~ 70% and no deformation ~15% of the time.

The subglacial processes derived from the study of contemporary glaciers may be useful in reconstructing the rate and nature of processes associated with Quaternary tills. Sliding may be associated with lodgement till[28], whilst the deformation recorded at our sites will result in deformation till[7,31,55]. Glacio-fluvial elements within tills may also be important indicators. Classically eskers are associated with channelized drainage[56], whilst the sedimentary remains of 'canals', 'subglacial meltwater corridors' and murtoos[50,57–59] may reflect the distributed system. Stratified lenses within till are common and have been given numerous interpretations, either reflecting preglacial sediments that have been incorporated into the till, usually by attenuation and boudinage[60] or penecontemporaneous sedimentation with the till either in a multichannel system[60,61] or at the ice

**Table 3 | Relative time periods for the different subglacial processes**

| Phase | Skálafellsjökull[a] | | | Fjallsjökull | | | Breiðamerkurjökull | | | Briksdalsbreen | | |
|---|---|---|---|---|---|---|---|---|---|---|---|---|
| | Melt season % | Winter % | Whole year % | Melt season % | Winter % | Whole year % | Melt season % | Winter % | Whole year % | Melt season % | Winter % | Whole year % |
| Phase 1/A –Sliding associated with the speed-up event | 18 | 9 | 13 | 17 | 11 | 14 | 29 | 10 | 19 | 15 | 6 | 12 |
| Phase 2/B - Deformation associated with reconnection as glacier slows down | 51 | 10 | 30 | 46 | 28 | 36 | 33 | 32 | 32 | 35 | 13 | 27 |
| Phase 3 Low deformation associated with velocity minimum | 17 | | 26 | 13 | | 23 | 13 | | 17 | 28 | | 41 |
| Phase C Stick | | 35 | | | 33 | | | 21 | | | 67 | |
| Phase 4/D - Deformation associated with reactivation as glacier begins to speed up | 14 | 46 | 31 | 25 | 28 | 27 | 25 | 37 | 31 | 22 | 14 | 19 |
| Annual Sliding | 13 | | | 14 | | | 19 | | | 12 | | |
| Deformation | 69 | | | 69 | | | 70 | | | 65 | | |
| No deformation | 18 | | | 17 | | | 11 | | | 23 | | |

[a]The value is different to that previously quoted[28] as those only included one year, whilst the figure above is a mean of two years.

**Fig. 6 | Typical Quaternary examples of stratified sand lenses within deformation till from West Runton, Norfolk, UK. a** Rounded and semi-angular sand lenses. **b** Deformed sand lenses. **c** folded sand lenses. **d** tear-drop shaped sand lens. These lenses represent either outwash sand deposition before the glacier advance and subsequently deformation or penecontemporaneous sedimentation and subsequent deformation within a multichannel system. Scale of sand lens in c equals 2.1 m x 1 m, and in d equals 0.5 m x 0.3 m.

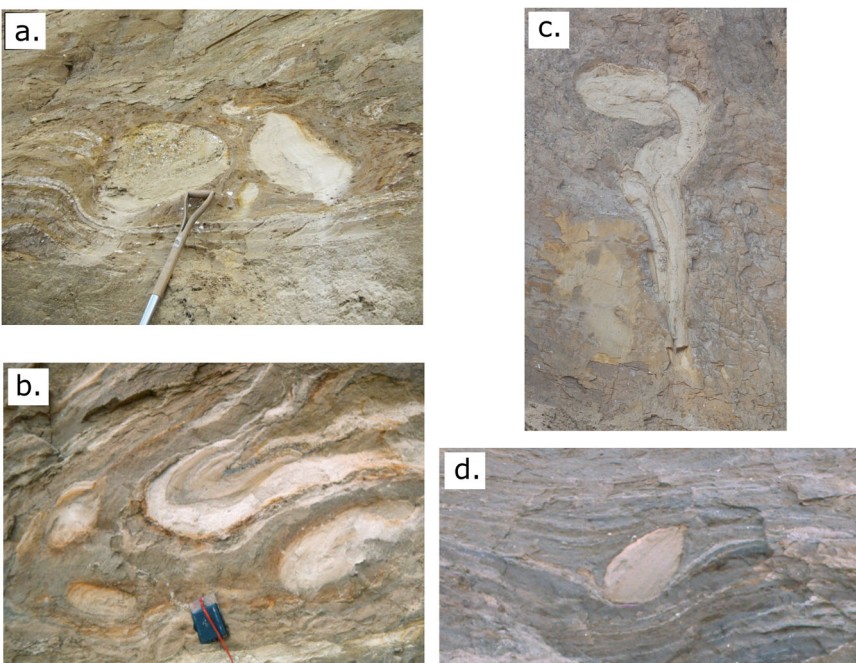

sediment interface[62,63]. These are very often subsequently deformed as they are deposited associated with a deforming bed. Our study highlights that we would expect to find sedimentary evidence for a distributed system in subglacial tills, and these would very likely be deformed given the high duration of deformation associated with a multichannel system (Fig. 6).

We have shown that there is a continuum between channelised and distributed drainage associated with soft bedded glaciers, with a distinctive sedimentary signature, which may be related to grain-size but is also influenced by subaqueous processes. We have proposed a new methodology to identify subglacial hydrology from Sentinel-1 SAR imagery, supported by in situ subglacial probe and daily GNSS data. This is important because distributed/multichannel drainage has a higher water pressure (and therefore, lower effective pressure), resulting in faster surface velocity and more ice loss. Recent modelling studies have consistently shown that changes in subglacial water pressure control basal sliding, which itself is a key factor in the response of glaciers to climate change[64]. Our analysis has the potential to be applied to a wide range of temperate glaciers to determine the relative occurrence of different subglacial hydrological systems associated with soft-bedded glaciers and enable a larger scale testing of the properties determining the channelised/distributed continuum. This would enhance our understanding of the subglacial environment and result in more accurate simulations of glacier dynamics and resultant sea level rise.

## Methods
### Environmental sensor network and the in-situ wireless probes
An environmental sensor network system was designed to collect the in-situ probe data, comprising sensor nodes and base stations, which are linked together by radio networks. Data was recorded from the probes at Briksdalsbreen (2004–2006) every four hours and at Skálafellsjökull initially every hour (2008–2010), and then every 15 min during 2012. Their data was transmitted from base stations via GPRS to a cloud server and hence to a sensor network server in the UK. Node data, along with differential GNSS recordings and meteorological data, were sent once a day to a mains powered computer (5 km at Briksdalsbreen, 16 km Skálafellsjökull), where it was forwarded to a web server in the UK[28,34].

The specific site location on the glacier was determined by the optimal depth at which the system can transmit data through the till and ice (50–80 m). These probes were deployed in the summers of 2004, 2005, 2008 and 2012, in a series of boreholes, which were drilled with a Kärcher HDS1000DE jet wash system. Once the boreholes were made, the glacier and till were examined using a custom-made CCD colour video camera with infrared LED illumination. If till was present, it was hydraulically excavated[16,65] by maintaining the jet at the bottom of the borehole for an extended period of time. The probes were then lowered into this space, enabling the till to subsequently close in around them. The depth of the probes (in the till) was estimated from video footage of the ice/till interface to be ~0.1–0.3 m at Briksdalsbreen and ~0.1 − 0.2 m at Skálafellsjökull beneath the glacier base.

These water pressure data were calibrated against the measured water depths in the borehole immediately after probe deployment. The glacier thickness ($H$) was determined from measuring the depth of the boreholes and comparing with the GNSS data of the glacier surface. Effective pressure $N$ is calculated as follows:

$$N = P_i - P_w$$

where $P_i$ = pressure of ice, and $P_w$ = water pressure.

$$P_i = p_i g H$$

Where $p_i$ = density of ice (910 kg m$^{-3}$), $g$ = gravity (9.8 m s$^{-2}$) and $H$ = ice thickness.

In addition, custom-made, low-power geophones were installed within boreholes to avoid surface seismic noise. The geophone nodes continually sampled the output of three orthogonal geophones but only data from seismic events were stored, held temporarily on a micro-SD card until they were retrieved by the base station. We used a 25 dB amplifier to provide sufficient signal with a bandpass pre-filter of 0.5–234 Hz, and a sampling rate of 512 Hz[35].

### Internet of things real time kinematic (rtk) global navigation satellite system (GNSS)
We designed and built a unique low-cost, internet connected (with real-time solutions) GNSS with which to measure movement in remote locations. This was installed at Fjallsjökull and Breiðamerkurjökull (2017–2020). The system comprises a base station, one or more rovers and a server receiving

**Table 4 | Processing parameters used in SNAP to produce velocity rasters of each glacier**

| Glacier | Grid Azimuth Spacing (pixels) | Grid Range Spacing (pixels) | Registration Window Width/Height | Max. Velocity (m d⁻¹) |
|---|---|---|---|---|
| Skálafellsjökull | 5 | 5 | 64 × 64 | 1.5 |
| Breiðamerkurjökull | 20 | 20 | 256 × 256 | 5 |
| Fjallsjökull | 10 | 10 | 128 × 128 | 4 |
| Briksdalsbreen | 15 | 15 | 64 × 64 | 2 |

the data. It was based on L1/L2 dGPS from Swift Navigation (Piksi Multi), which use 3 W when operating, providing a typical accuracy of 2 cm after 40 s fix time and receive corrections from GPS, GLONASS and Galileo satellites. They are only powered on when taking readings to save battery lifetime.

The system was controlled by an ARM M4 microcontroller (96 kB RAM, 384 kB flash), which has a sleep current of 6 μA and ran micropython. Synchronised base station units were placed in the foreland, which transmitted the GNSS corrections to the rovers according to the schedule. An Iridium short messaging unit (Rockblock) was used by the rovers to send 8 readings once per day (330 bytes) directly to our database, allowing daily updates of data interpretation.

The annual data were collected by Version 1 of the system which operated 2017–2020. The data shown is from Fjallsjökull, located where the ice was ~77 m deep, and Breiðamerkurjökull where the ice was 84 m thick. The summer diurnal data were collected from Version 2 of the system installed in 2024, which was installed in a similar location.

### Velocity records

Ice surface velocity was measured at Skálafellsjökull from 2008–2012 with a TOPCON Legacy-H L1/L2 GPS (1 km baseline), and from 2012-2013 with an additional array of 4 dual frequency Leica 1200 GPS systems which obtained data continuously during the summer and 2 h a day during the winter at a 15 s sampling rate (300 m baseline). The GPS data were then processed using data from the International GPS Service (IGS) reference stations using TRACK (v. 1.24), the kinematic software package developed by Massachusetts Institute of Technology (MIT) http://geoweb.mit.edu/~tah/track_example/). We derived an average surface horizontal velocity by taking the mean of 4 GPS stations to remove local variations. To account for surface melting, we removed the daily melt from the vertical measurements. The mean error estimates were as follows (sigma per day): North +/− 0.0045 m, east +/− 0.0032 m, height +/− 0.0092 m. We have previously demonstrated[16] that velocity has a distinct pattern related to air temperature and utilised a transfer function to reconstruct a velocity record for 2009/10 using the velocity data from 2012/13.

At Fjallsjökull and Breiðamerkurjökull, the surface velocity was measured with the web connected RTK GNSS system discussed above. The error estimates were +/- 2 cm. At Briksdalsbreen the surface velocity was measured with a TOPCON legacy L1/L2 GPS (1 km baseline), however the data could only be calculated on a quasi-monthly timescale. A monthly mean was taken from the two-year record and scaled to the beginning of the record. The error estimates were +/- 2 cm.

Sentinel-1 SAR imagery, obtained in IW mode, was also used to calculate glacier-wide velocities at 12-day repeat intervals. Data were generated using the offset tracking algorithm within the European Space Agency (ESA) Sentinel Application Platform (SNAP). Although offset tracking is less precise than SAR interferometry, given the high temporal correlation of glacier surfaces, it is much more robust, and as such the method is widely used in glacier motion assessment[66]. Here, each pair of SAR images were first radiometrically calibrated and then co-registered using the aerial LiDAR DEM of Iceland, provided at 10 m resolution by the National Land Survey of Iceland. Velocities were then calculated using cross correlation, with specific parameters, including the moving window size and search distance, varying between each glacier (Table 4).

Any displacements with a cross-correlation threshold of <0.01 were then removed, with the remaining displacements averaged over a mean pixel grid and converted to ground range coordinates, resulting in velocity rasters at 10 m resolution. Mean velocities were then calculated along the centre line of each glacier. The stochastic error in our velocity measurements was assessed by measuring displacements over terrain that we regarded as stable[67,68]. The average RMSE for the Sentinel-1 imagery over the entire period was +/−0.15 m per day. This was calculated from June 2017 to Oct 2020.

We calculated the change in 12-day velocity over the summer by comparing the maximum velocity during the spring with the lowest velocity just prior to the autumn velocity rise, which we express as a percentage. We do this for each year, taking a mean for each glacier.

### Air temperature records, defining the seasons and determining the relationship between winter events and velocity change

Air temperature data were primarily obtained from the base stations described above but during periods of mechanical failure, a transfer function was applied to data from neighbouring meteorological stations. For Skálafellsjökull we used Hofn (~30 km away), for Fjallsjökull and Breiðamerkurjökull we used Kvisker (~6 and ~16 km away respectively) and for Briksdalsbreen we used Stryn (~30 km away).

We define the seasons based on air temperatures, and since the sites have different mean annual air temperatures (Table 1) we utilised slightly different seasonal thresholds. Winter is a time where there is little melting as air temperatures are low. We have used the mean annual air temperature as this threshold (Skálafellsjökull 0 °C, Fjallsjökull and Breiðamerkurjökull 5 °C, and Briksdalsbreen 6 °C). Spring begins once the daily air temperatures are continuously above the winter threshold, summer is reflected by much higher air temperatures (~5 °C higher than the winter threshold), autumn air temperatures are lower often falling below zero at night.

We quantified the glacier response to the winter events in two ways. For the daily velocity data from GNSS, we counted the percentage of events where the speed-up event occurred at the same time as a temperature rise above the threshold. For the 12-day velocity data from Sentinel-1, we calculated the number of days within each period that air temperatures were above the threshold ($D_N$). We assumed that if the value of $D_N$ was zero, it would reflect a base (low) velocity, if it were above zero it would reflect a speed-up event (high velocity). We then calculated the percentage of 'correct' attributions over the winter. We set the threshold between low and high velocity at 95% of the maximum winter velocity.

### Determining the different phases of stick-slip motion

The initial determination of the different phases of stick-slip motion were calculated from the tilt, geophone, velocity and air temperature data from Skálafellsjökull. There are four phases (diurnal in melt season indicated by a number, multi-day in winter by a letter): sliding (Phase 1/A), reconnection (Phase 2/B), low or no deformation (Phase 3/C), reactivation (Phase 4/D). We were able to extend this analysis to the other glaciers in the study. At Fjallsjökull and Breiðamerkurjökull, although we do not have tilt or geophone data, the data from Skálafellsjökull clearly illustrate a relationship between tilt and velocity, so that velocity can be used as proxy for tilt in the analyses. Although the velocities from Skálafellsjökull were very similar during Stages C and D, they could be distinguished using air temperature: Stage C occurred when air temperatures were low (below zero), whilst Stage

D was associated with rising air temperatures (just before the threshold for the winter event to occur). This enabled us to isolate the two stages at Fjallsjökull and Breiðamerkurjökull: Stage C was associated with low velocity and low air temperatures (less than 5 °C), and Stage D with rising air temperatures (and occasionally higher velocity).

At Briksdalsbreen, where tilt data was available but not velocity, it was possible to use these alongside air temperatures to identify the different phases. During the melt season, there was a distinct daily pattern. As air temperatures (and presumably velocities) rise in the morning they are accompanied at midday by low tilt change, which we suggest reflects sliding (Phase 1). During the afternoon, there was an increase in tilt change, which we suggest reflects reconnection with the bed (Phase 2) and deformation throughout the afternoon and evening. Tilt motion/deformation is then lowest at midnight (Phase 3), before increasing slightly throughout the morning as the glaciers speeds up in response to increasing melt (Phase 4). During winter, there was also a pattern of stick-slip motion but with a more complex, double configuration. When air temperatures drop below zero, water pressures rise, until a water pressure threshold is crossed (after ~four days), and we assume there is a speed-up event that generates the antithetic behaviour and sliding (Phase A) (Fig. 4a, DOY 23). After this the glacier slows down and reconnects with the bed, resulting in deformation (indicated by the large synthetic tilt movement) (Phase B) (Fig. 4a, DOY 24). If the water pressure is sufficiently high then there will not be a stick phase, but rather continuous low synthetic deformation. This is often indistinguishable from the reconnection Stage D (Fig. 4a, DOY 25-34) so we have grouped the two together. Then as air temperatures again rise above zero we see a pattern similar to Skálafellsjökull. Firstly, meltwater enters the system, resulting in sliding and antithetic deformation (Phase A) (Fig. 4a, DOY 35), then a dramatic decline in water pressure associated with synthetic tilt (Phase B) (Fig. 4a, DOY 36), followed by a period of stick (Phase C) (seen in event 1) or low deformation (Phase D) (Fig. 4a, DOY 37-42) (seen in the rest of the events) depending on porewater pressures.

## Data availability
Data is available at Glacsweb.org (https://data.glacsweb.info/datasets/) and (https://doi.org/10.5258/SOTON/D3403).

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

## Acknowledgements

The authors would like to thank the Glacsweb 2003–2024 teams for help with probe development and data collection. They would also like to thank Matthew Roberts of the Icelandic Meteorological office for his advice and support. We would also like to thank Dr Phillip Basford and Josh Curry and for help with design of the Smart tracker and database. The authors also thank Eyjólfur Magnússon for sharing his bedrock topography data for Fjallsjökull. This research was funded by EPSRC (EP/C511050/1), NERC (NE/L012405/1), Leverhulme (F/00180/AK, RPG-2021-316) and the National Geographic (GEFNE45-12, NGS-368R-18) and the GPR and Leica 1200 GPS units were loaned from the NERC Geophysical Equipment Facility.

## Author contributions

J.K.H. and K.M. designed the study. J.K.H. carried out the probe and GNSS data analysis. K.M. and G.B. designed the sensor network system, Glacsweb probes and web connected GNSS system as well as the software. N.R.B., B.A.R., A.B derived the remotely sensed surface velocity. J.K.H. wrote the manuscript with input from all authors.

## Competing interests

The authors declare no competing interests.
