## [Transparent Peer Review file · Communications Earth & Environment]

Seasonal velocity patterns provide insights for the soft-bed subglacial hydrology continuum

Corresponding Author: Professor Jane Hart

Version 0:

Decision Letter:

Dear Professor Hart,

Your manuscript titled "Seasonal velocity patterns and deforming bed processes associated with different subglacial drainage systems." has now been seen by 2 reviewers, and we include their comments at the end of this message. They find your work of interest, but some important points are raised. We are interested in the possibility of publishing your study in Communications Earth & Environment, but would like to consider your responses to these concerns and assess a revised manuscript before we make a final decision on publication.

We therefore invite you to revise and resubmit your manuscript, along with a point-by-point response that takes into account the points raised. Specifically, we ask you to perform further analysis of satellite data to better verify flow dynamics, and better explain the global implication of the results as well as the impact of categorization of glaciers based on physical processes on reconstructions and projections. Please highlight all changes in the manuscript text file.

Please submit your point-by-point responses as a separate file, distinct from your cover letter where you can add responses to the Editors' comments that you do not want to be made available to the reviewers. Word files are preferred.

Important: The response to reviewers must not include any figures, tables or graphs. If you wish to respond to the reviewer reports with additional data in one of these formats, please add them to the main article or Supplementary Information, and refer to them in the rebuttal. Due to current technical limitations, any figures, tables, or graphs embedded in your rebuttal will not be included in the peer review file, if published.

Please use the following link to submit your revised manuscript, point-by-point response to the referees' comments (which should be in a separate document to any cover letter), a tracked-changes version of the manuscript (as a PDF file) and the completed checklist:

Link Redacted

We hope to receive your revised paper within six weeks; please let us know if you aren't able to submit it within this time so that we can discuss how best to proceed. If we don't hear from you, and the revision process takes significantly longer, we may close your file. In this event, we will still be happy to reconsider your paper at a later date, as long as nothing similar has been accepted for publication at Communications Earth & Environment or published elsewhere in the meantime.

Please do not hesitate to contact us if you have any questions or would like to discuss these revisions further. We look forward to seeing the revised manuscript and thank you for the opportunity to review your work.

Best regards,

Dr Alireza Bahadori
Associate Editor
Communications Earth & Environment

EDITORIAL POLICIES AND FORMATTING

Editorial Policy: [Policy requirements](https://www.nature.com/documents/nr-editorial-policy-checklist.pdf) (Download the link to your computer as a PDF.)

- Behavioural and social science
- Ecological, evolutionary & environmental sciences
- Life sciences

<https://www.nature.com/documents/nr-reporting-summary.zip>

Furthermore, please align your manuscript with our format requirements, which are summarized on the following checklist: [Communications Earth & Environment formatting checklist](https://www.nature.com/documents/commsj-phys-style-formatting-checklist-article.pdf)

and also in our style and formatting guide [Communications Earth & Environment formatting guide](https://www.nature.com/documents/commsj-phys-style-formatting-guide-accept.pdf) .

*** DATA: Communications Earth & Environment endorses the principles of the Enabling FAIR data project (<http://www.copdess.org/enabling-fair-data-project/>). We ask authors to make the data that support their conclusions available in permanent, publically accessible data repositories. (Please contact the editor if you are unable to make your data available).

All Communications Earth & Environment manuscripts must include a section titled "Data Availability" at the end of the Methods section or main text (if no Methods). More information on this policy, is available at <http://www.nature.com/authors/policies/data/data-availability-statements-data-citations.pdf>.

If a community resource is unavailable, data can be submitted to generalist repositories such as [figshare](https://figshare.com/) or [Dryad Digital Repository](http://datadryad.org/). Please provide a unique identifier for the data (for example a DOI or a permanent URL) in the data availability statement, if possible. If the repository does not provide identifiers, we encourage authors to supply the search terms that will return the data. For data that have been obtained from publically available sources, please provide a URL and the specific data product name in the data availability statement. Data with a DOI should be further cited in the methods reference section.

REVIEWER COMMENTS:

Reviewer #1 (Remarks to the Author):

This manuscript puts forward an analysis of data collected from 4 glaciers that are analysed to understand subglacial hydrology organisation and bed deforming processes. I find the paper interesting and novel with potentially a significant outcome in the attempt to improve our ability to characterise these systems that are very hard to observe in-situ. I think the manuscript would benefit from resolving some unclearness and an increase in the precision

From the abstract the major points made seem to be 1) there is a definable and measurable difference in flow dynamics between braided and channelised systems, for which criteria are presented, 2) these differences relate to bed processes enabled by differences in the subglacial sediments, 3) the seasonal velocity pattern of these subglacial regimes is different, providing a basis for classification and 4) that these techniques are valuable for the broader scale prediction of glacier behaviour and sea level rise.

With respect to 1) for the four glaciers discussed, their differences are well measured and described, however, the observation period (one year) and the low number of glaciers leads to a significant uncertainty - how representative is the data? Also, the matter of flow-network topology is fundamentally a 2D + time problem - but the data are 0D + time...it was not clear to me how the 'spot' data from the field sites map to defining regional network-scale behaviour of the system described in the discussion. I would like to see either a longer-term 2D analysis of the satellite data, or some sort of physical model to verify the interpretation.

with respect to 2) I find this point intriguing but not very clearly made. Yes there is a difference in grain size between the sites, and a comment is made concerning the expected type of till that might be associated with the different regimes. But the evidence provided is far from clear-cut and there is a complex set of factors to consider - for example sediment transport capacity, porosity, permeability etc. For me I think this point is not central to achieving the desired impact

for 3) some criteria are defined for the segregation of braided and channelised beds from seasonal velocity patterns. This is qualitative rather than quantitative but is a good starting point to discuss how this might be realised at a global scale...this comes back to representativeness of the data set, but also to the uniqueness of the velocity signal.

The unfortunate fact that for the channelised flow the velocity data are from a different dataset is an Achilles heel - how do we know that Nigardsbreen is not responding completely differently to Brikdalsbreen? I think there needs to be a clear basis for this substitution as valid before the outcomes can be relied upon

Finally for 4) I think there is a need to make a stronger connection between the rather specific outcomes here and the global problem...what is the path for uptake? there needs to be some connection to how the processes might be incorporated in modelling and/or global-scale accounting of glacial mass balance

General points:

The figures have inconsistent formatting and scales, including changing units and are somewhat hard to digest. If these could be made more consistent it would be easier to follow the text

The discussion is quite long and not very well structured - I found it hard to follow the thread. I think it needs broken up into subsections and a little more care given to the logical structure of the arguments made.

I think the concluding paragraph is underwhelming. It needs to have a much firmer statement around what the outcome(s) of the paper are and their importance for a general audience.

Specific comments:

Comments are attached in a PDF

Reviewer #3 (Remarks to the Author):

In this study, the authors present a long term comparison of ice velocity, with basal water pressure, air temperature, and other constraints dealing with the geologic setting of each glacier, to investigate the interaction of climate and glacier flow, and highlight different groups of subglacial drainage systems and their characteristic response in terms of ice velocity. The data set is impressive and the conclusions reached by the authors are quite meaningful.

I have a few comments about the organization of the paper and a few comments on the impact of the results. Overall, I am very supportive of publication of the paper with minor revision.

Abstract: Could be improved. I do not like the parenthesis in the first sentence of the abstract, please omit them and make nice sentences instead. Explain "remotely sensed" as being satellite data from Sentinel-1 (otherwise too vague). The word "argue" seems out of place. The data shows, the study reveals, no need to argue. "we are able" are able do not add anything to the sentence. The sentence "This is important .." the "this" applies to what? This alone is confusing. Better to say "This categorization of glacier ..." Overall the abstract is poor and needs to be better written. The last sentence is a bit of a motherhood statement. I am not sure in particular that the particular subglacial hydrology regime is key to projections of sea ice rise. You say that, but you do not show that in the paper and I do not think it is correct generally speaking.

Introduction. A lot of what is written applies to land terminating glaciers or marine terminating glaciers? In the case of marine terminating, it is too simplistic to only mention subglacial hydrology and ignore for instance the role of ice-ocean interactions, which are by far dominant.

Line 35. Most of what is said here, and the paper, applies to glaciers in warm environments. There is no summer melt cycle in Antarctica for instance. So the impact of this study on modeling Antarctic glaciers is probably minimal. Please clarify that this study applies to glaciers with strong seasonality and summer melt.

Line 60-78. This passage absolutely does not belong there. These lines are a summary of the paper, which do not belong in the introduction. Please correct accordingly. Introduce your study, method, and that we will draw conclusions on the impact of subglacial hydrology on glacier response, but do not give us the results. This is the introduction, never a summary of the results and discussion.

Line 79. Glacsweb is some sort of acronym. It does not describe the instruments being used. Please refer to the materials and methods and describe these instruments. I went into the literature of Glacsweb to figure out the details, and got lost, and annoyed, I did not find what I wanted easily.

General comments: "We can" is to be avoided. Either "we do " or "we did". We can means you could do that, but did you really? It makes the sentence weak.

Second one. Please avoid sentences with "This means, this is" because I do not know what "this" relates to. There needs to be a noun.

Figure 2. The vertical labels are too small to read. Very painful. Make the labeling shorter and 3 x times larger letters. Include minor tickmarks on a daily basis.

No vertical error bars?

Line 245. Again, Svalbard glaciers vs Antarctic glaciers? No summer melt in the latter. Please place in proper context.

The rest of the discussion is good, albeit a bit long.

Line 429. Reference for first sentence?

Generally speaking, keep the present tense for describing your results, and the past for already published material, otherwise the reader is confused.

Line 446. "WE have proposed"  "We propose "

Line 447. I still do not know Glasweb. May be water pressure and air temperature probes?

Now on broader terms, you identify different glacier regimes in response to different subglacier hydrology regimes. That's great because it is practically impossible to sample the subglacial hydrology. To make the point that this matters, I would ask the following: have you documented on how a particular subglacial regime affects projections of mass loss vs another one? In other words, is it well known how the selection of a subglacial regime will affect the projections of sea level rise? More important, can you demonstrate that if you make a mistake, you cannot replicate the glacier evolution, whereas if you make the right choice, it will be easy to reconstruct the glacier evolution? My point is that the impact of the results may be a bit oversold. I agree that this is a very elegant way to categorize the glaciers, based on physical processes. It is less well clear how much this will affect reconstructions and projections. My two cents.

This paper remains an impressive examples of dense observation network on a set of glaciers, combined with satellite remote sensing.

Communications Earth & Environment is committed to improving transparency in authorship. As part of our efforts in this direction, we are now requesting that all authors identified as 'corresponding author' create and link their Open Researcher and Contributor Identifier (ORCID) with their account on the Manuscript Tracking System prior to acceptance. ORCID helps the scientific community achieve unambiguous attribution of all scholarly contributions. You can create and link your ORCID from the home page of the Manuscript Tracking System by clicking on 'Modify my Springer Nature account' and following the

instructions in the link below. Please also inform all co-authors that they can add their ORCID to their accounts and that they must do so prior to acceptance.

Version 1:

Decision Letter:

Dear Professor Hart,

Your revised manuscript titled "Seasonal velocity patterns and deforming bed processes associated with different subglacial drainage systems." has now been seen by our original reviewer 3 and a new reviewer 4 who replaces the original reviewer 1 (reviewer 1 was not available for further comments). All comments appear below. In light of their advice we are delighted to say that we are happy, in principle, to publish a suitably revised version in Communications Earth & Environment, provided you include a more thorough discussion of the implications of subglacial hydrology classification on glacier modeling and projections.

We therefore invite you to revise your paper one last time to address the remaining concerns of our reviewers. At the same time we ask that you edit your manuscript to comply with our format requirements and to maximise the accessibility and therefore the impact of your work.

EDITORIAL REQUESTS:

****Please take care to match our formatting and policy requirements. We will check revised manuscript and return manuscripts that do not comply. Such requests will lead to delays. ****

SUBMISSION INFORMATION:

OPEN ACCESS:

Communications Earth & Environment is a fully open access journal. Articles are made freely accessible on publication. For further information about article processing charges, open access funding, and advice and support from Nature Research, please visit <https://www.nature.com/commsenv/open-access>

Link Redacted

Best regards,

Alireza Bahadori, PhD
Associate Editor
Communications Earth & Environment

REVIEWERS' COMMENTS:

Reviewer #3 (Remarks to the Author):

Thank you for the revision. Generally speaking, it is not helpful for a reviewer to read in the rebuttal letter a statement that says "we made some changes". I do not know what changes are made and where you implemented it. So you are implicitly asking me to go find out. It would be more efficient to indicate what you change, why, and where, e.g. a line number in the MS.

The abstract has a mix of sentences in the present tense (correct) and past (not good to change style). I mentioned that issue before, but you only changed it in a few places. Either you ignore my recommendation, or you implement it in full, including the abstract. I suspect the co-authors of the paper, who include senior authors, could help by reading the rebuttal and revised manuscript. The fact that these changes did not occur suggest that only one author worked on the revision. It would be a good mentoring to help the first author.

Line 27-28. Add freshwater resources. Mountain glaciers matter for sea level, but more so in terms of freshwater resources to local populations in the Andes and Himalayas.

Line 55. Lower pressure, lower basal friction, more speed and faster retreat. Is that so? When effective pressure is high, ice is compressed against the bed, and friction is low. When the pressure is low, ice is less pressed against the bed and friction is lower. As water pressure increases, the effective pressure decreases, and basal friction is lower. So are you saying "water pressure" or "effective water pressure"? One P_{water} , the other is $P_{\text{ice}} - P_{\text{water}}$. This is better worded at line 457. Please make sure you say it right because this sentence has major implications for your work.

Introduction: You do not want to mention ocean-ice interaction as a key control in addition to subglacial hydrology and likely more important because "it is beyond the scope of your paper"? Well, this is common knowledge. It is not a matter of changing your scope, it is a matter of placing your study into a broader context. If you do not agree, that is one thing. If you say it is beyond the scope, I do not agree with you. Breiðamerkurjökull seems to be a tidewater glacier (line 314, is that correct? The image in figure 1 shows no obvious connection with the ocean) while the others are lake terminating or land terminating, so ocean-ice interaction has an impact on its evolution that you must consider (e.g., migration of the line of grounding, seasonal melt along the ice face depending on subglacial water plume and ocean temperature, tidal-modulation of the horizontal speed, etc.). If ending only in a freshwater lake, there are still water-ice interactions that matter. In fact, you say so at line 300 ..

Discussion. Present tense would make it better - again.

Results: Describe your new work with the present tense. Published work can be in the past tense. This approach will make your text livelier. Do it in the methods as well. You can choose not to do it, of course, your decision, but you cannot mix styles within the same paragraph.

Last comment: "the modeling is beyond the scope of the paper". Well, this is related to the broader impact of your study. You need to discuss how the classification of glacier into different subglacial regimes might affect – or not – glacier modeling and/or projections. I am not asking you to conduct extra work here, e.g., a specific modeling study, but rather to cite literature or past work that would indicate how impactful this will be. If this classification will impact model projections at the 0% or 100% level, it is good to know, on a best effort basis, to understand why you need to distinguish these regimes in the first place. At face value, Breiðamerkurjökull differs from the others due to the tidally connected lake. Perhaps that connectivity matters more than you think .. Overall, my burning question remains: why should we care about the type of subglacial hydrology if it does not impact projections?

Data availability: Good job with the web site and data access/sharing. Would be nice to get a DOI at the time of submission.

Figures are much improved. I am not 100% sure about having both water pressure and overburden pressure. Perhaps the latter is most relevant.

Minor ones:

Line 116. What do you mean by averaging velocity over the whole glacier? Do you mean across the glacier width? At what location do you average the satellite-derived velocity w.r. to the ELA? At the ELA (max speed)? At the calving front – when there is max speed if calving into a lake otherwise you get zero. The text says "along the center line", which I find puzzling. I would prefer ELA or calving front.

Line 340. Typo 50

This paper would benefit from another round of revisions

Reviewer #4 (Remarks to the Author):

Dear Jane and co-authors,
thanks a lot for this interesting paper. The study compares the glacier velocity behaviour (seasonal and diurnal) of four different glaciers located in Norway and Island. The measurements of glacier velocity (GNSS and from satellite data), meteorological data (air temperature), water pressure, ice thickness, and till grain size were visually compared to categorized the subglacial hydrology into braided, drained braided, and channelized drainage system. The annual process of the four glaciers are very detailed and comprehensive explained and combined with the water drainage and storage. This paper is very valuable for the understanding of subglacial hydrology and provides new insights into the processes regarding hydrology, bed deformation, and glacier velocity. There are rarely as many parameters available as here to visualise the entire process in such detail. I would address some concerns and remarks. It would be great if you could include them in the newer version to make your manuscript better readable and understandable and on the other side to strengthen your statements:

1. The abstract starts with the measurements and not with the background of the problem or why this study is important. Your list all measurements, but it is not clear which parameter are exactly measured. Additionally, the results or the categorization of the drainage is not cleared explained (which types exists and where are the differences) and especially how satellite data could use to recognize the different hydrology types is not well described. Please, revise the abstract including my three above mentioned points. Avoiding repetitions at the beginning of sentences (first three sentences start with 'we') would improve the abstract immense.
2. Table 1 indicated that the parameter like the velocity is available for five years, but the plots present just one year. Is there a reason why you selected just a specific year and showing all the other years? Additionally, some statistics would be essential to strengthen your observations.
3. In the introduction, you mention the distributed and the channelized drainage. But in Figure 5, just the braided, drained braided, and channelized drainage are displayed and mentioned in the discussion. Is the distributed drainage not relevant in this categorization, is it replaced by the term 'braided', or is it just relevant for hard glacier bed? For me it was little bit confusing and it would be great to explain all these definitions in more detail in the introduction. This would be very helpful for the reader and enhance the comprehension – as well if this study focuses only on soft glacier beds (if yes you could add this in the title) and what the difference between continuum and separated drainage is.

Minor corrections:

- Line 28: change to "The higher air temperature". If you could change temperature to air temperature, the reader knows that you mean the air temperature and not eventually the glacier temperature. Please take care of it in the complete manuscript.
- Line 58: Repetitions at the beginning of sentences ('We').
- Line 66-68: Do you have a reference to prove this?
- Line 72: Change to "lower water storage"
- Line 69-74: Do you have a reference to prove this?
- Line 111: Do you mean 2017/18?
- Line 117: Was the surface velocity derived from the satellite data or using GNSS?
- Line 125-126: But in Figure 2a, water pressure increases as well as velocity during summer? I have the feeling that plot and description does not match or am I wrong?
- Line 126: repetition of also (used twice in one sentence).
- Line 148: What causes the ice-quakes?
- Line 149: How can the GPR recognise that it is a braided subglacial hydrology?
- Line 182-192: Please could you refer the corresponding figures? I assume you refer to Figure 2b and 4b?
- Line 188: Do you mean that the air temperature rose above the spring temperatures? This sentence is not clear for me.
- Line 198: Do you mean Figure 2c?
- Line 210: What means the numbers of the probe? Are there relevant for the reader? In Line 234 you are using only the number without the 'B'.
- Line 212: Please could you indicate the date to the DOY like you did in Line 186? That would make it much easier for the reader to sort the events in the time.
- Line 224: Can you explain the reason for that double event?
- Line 226: Spring in lowercase.
- Line 244: Cannot find Figure 3d.
- Line 258: You mention the distributed drainage, but it is not displayed in Figure 5.
- Line 268: Could you give a reason for the velocity decline?
- Line 305 and 340: Reference superscript.
- Line 369: Could you indicate the grain size in numbers?
- Line 425 – 430: Would make sense to add phase in brackets to make it easier for the reader to combine text and table.
- Line 438: What are modern glaciers?
- Line 455: An explanation if the temporal resolution of S1 is sufficient to categorize the glacier into the different hydrology systems would be great!
- Line 538: Which mode have you used?
- Line 543: Which steps include the calibration (geometric and/or radiometric)?
- Line 554: Could you add a sentence how you calculated the RMSE, please?

Figure 2: Colours of air temperature and velocity can hardly be distinguished. Indicate the complete time period with start

and end. The plot is slightly overcrowded and it takes time to understand all the information. Perhaps the plots could be slightly improved graphically to make them more readable (same for Figure 3).

Figure 3: Colours of air temperature and velocity can hardly be distinguished. I assume that the velocity was derived from satellite data, hence it would make sense to add this information in the caption. Indicate the complete time period with start and end.

Figure 4e: I assume you mean DOY 19-42 as mentioned in the text.

We thank the referees for their helpful comments. We have reflected on these and improved the manuscript accordingly. Please find enclosed our point-by-point response to the referees' comments. The referees' comments are in italics and our replies are in normal text. We have also included our responses to the comments on the marked-up pdf.

Reviewer comments:

Reviewer #1 (Remarks to the Author):

This manuscript puts forward an analysis of data collected from 4 glaciers that are analysed to understand subglacial hydrology organisation and bed deforming processes. I find the paper interesting and novel with potentially a significant outcome in the attempt to improve our ability to characterise these systems that are very hard to observe in-situ. I think the manuscript would benefit from resolving some unclarity and an increase in the precision

From the abstract the major points made seem to be 1) there is a definable and measurable difference in flow dynamics between braided and channelised systems, for which criteria are presented, 2) these differences relate to bed processes enabled by differences in the subglacial sediments, 3) the seasonal velocity pattern of these subglacial regimes is different, providing a basis for classification and 4) that these techniques are valuable for the broader scale prediction of glacier behaviour and sea level rise.

With respect to 1) for the four glaciers discussed, their differences are well measured and described, however, the observation period (one year) and the low number of glaciers leads to a significant uncertainty - how representative is the data? Also, the matter of flow-network topology is fundamentally a 2D + time problem - but the data are 0D + time...it was not clear to me how the 'spot' data from the field sites map to defining regional network-scale behaviour of the system described in the discussion. I would like to see either a longer-term 2D analysis of the satellite data, or some sort of physical model to verify the interpretation.

Our response: Collecting any subglacial data is very difficult. Although we accept that 4 glaciers is a small number, this provides the basis for other studies. The time for each glacier varies but was longer than 1 year between 5 to 6 years (see Table 1). We link together the detailed studies from the wireless probes, GNSS and satellite data. Sentinel-1 data is only available since 2017. In future we could investigate more glaciers by remote sensing, but we specially wanted to 'ground truth' them with field data.

with respect to 2) I find this point intriguing but not very clearly made. Yes there is a difference in grain size between the sites, and a comment is made concerning the expected type of till that might be associated with the different regimes. But the evidence provided is far from clear-cut and there is a complex set of factors to consider - for example sediment transport capacity, porosity, permeability etc. For me I think this point is not central to achieving the desired impact

Our response: We welcome this point and have added more detail. Other researchers are working on the sliding law issue from a theoretical point of view and our results can be used by them.

for 3) some criteria are defined for the segregation of braided and channelised beds from seasonal velocity patterns. This is qualitative rather than quantitative but is a good starting point to discuss how this might be realised at a global scale...this comes back to representativeness of the data set, but also to the uniqueness of the velocity signal.

Our response: This relates to point 1. Although they are only four well observed soft bedded glaciers they do appear to have different hydrologies. Studies of soft-bedded glaciers are very rare, although vital for our understanding of numerous modern day and Quaternary glaciers. Our model is based on ground truthing. The next step is to apply this globally which would be beyond the scope of paper (we are also at the word limit already). Our plan is to use these ground truthed sites and then go on to examine this more globally.

The unfortunate fact that for the channelised flow the velocity data are from a different dataset is an Achilles heel - how do we know that Nigardsbreen is not responding completely differently to Briksdalsbreen? I think there needs to be a clear basis for this substitution as valid before the outcomes can be relied upon

Our response: We accept that they are different glaciers but Sentinel-1 was not in operation when Briksdalsbreen was studied (and it has now retreated almost up to the plateau). We have now provided more evidence that they are sufficiently similar to be compared, and added our GPS data from Briksdalsbreen at a monthly scale.

Finally for 4) I think there is a need to make a stronger connection between the rather specific outcomes here and the global problem...what is the path for uptake? there needs to be some connection to how the processes might be incorporated in modelling and/or global-scale accounting of glacial mass balance

Our response: We have taken this comment on-board and have now expressed this more clearly.

General points:

The figures have inconsistent formatting and scales, including changing units and are somewhat hard to digest. If these could be made more consistent it would be easier to follow the text

Our response: We have addressed this point.

The discussion is quite long and not very well structured - I found it hard to follow the thread. I think it needs broken up into subsections and a little more care given to the logical structure of the arguments made.

Our response: This journal does not allow subsections in the Discussion. However, we have made improvements to the logical structure.

I think the concluding paragraph is underwhelming. It needs to have a much firmer statement around what the outcome(s) of the paper are and their importance for a general audience.

Our response: We have now improved this.

Specific comments:

Comments are attached in a PDF

Our response: These are all addressed. The point about mean grain size, we changed to a qualitative description to capture the range of grain sizes.

Reviewer #3 (Remarks to the Author):

In this study, the authors present a long term comparison of ice velocity, with basal water pressure, air temperature, and other constraints dealing with the geologic setting of each glacier, to investigate the interaction of climate and glacier flow, and highlight different groups of subglacial drainage systems and their characteristic response in terms of ice velocity. The data set is impressive and the conclusions reached by the authors are quite meaningful.

I have a few comments about the organization of the paper and a few comments on the impact of the results. Overall, I am very supportive of publication of the paper with minor revision.

Abstract: Could be improved. I do not like the parenthesis in the first sentence of the abstract, please omit them and make nice sentences instead. Explain "remotely sensed" as being satellite data from Sentinel-1 (otherwise too vague). The word "argue" seems out of place. The data shows, the study reveals, no need to argue. "we are able" are able do not add anything to the sentence. The sentence "This is important .." the "this" applies to what? This alone is confusing. Better to say "This categorization of glacier ..." Overall the abstract is poor and needs to be better written. The last sentence is a bit of a motherhood statement. I am not sure in particular that the particular subglacial hydrology regime is key to projections of sea ice rise. You say that, but you do not show that in the paper and I do not think it is correct generally speaking.

Our response: We have taken these comments on board and have improved the abstract.

Introduction. A lot of what is written applies to land terminating glaciers or marine terminating glaciers? In the case of marine terminating, it is too simplistic to only mention subglacial hydrology and ignore for instance the role of ice-ocean interactions, which are by far dominant.

Our response: We do mention that other factors are important, but here we concentrate on the subglacial hydrology. It is beyond the scope (and word limit) of the paper to discuss other factors.

Line 35. Most of what is said here, and the paper, applies to glaciers in warm environments. There is no summer melt cycle in Antarctica for instance. So the impact of this study on modeling Antarctic glaciers is probably minimal. Please clarify that this study applies to glaciers with strong seasonality and summer melt.

Our response: This is a good point so we have added this

Line 60-78. This passage absolutely does not belong there. These lines are a summary of the paper, which do not belong in the introduction. Please correct accordingly. Introduce your study, method, and that we will draw conclusions on the impact of subglacial hydrology on glacier response, but do not give us the results. This is the introduction, never a summary of the results and discussion.

Our response: This is the style of the journal , the instructions state “The final paragraph should be a brief summary of the major results and conclusions.”

Line 79. Glacsweb is some sort of acronym. It does not describe the instruments being used. Please refer to the materials and methods and describe these instruments. I went into the literature of Glacsweb to figure out the details, and got lost, and annoyed, I did not find what I wanted easily.

Our response: We have changed the reference to one more relevant, and avoided any acronyms.

General comments: "We can" is to be avoided. Either "we do " or "we did". We can means you could do that, but did you really? It makes the sentence weak.

Our Response: These are now all changed as suggested.

Second one. Please avoid sentences with "This means, this is" because I do not know what "this" relates to. There needs to be a noun.

Our response: We have gone through and checked all the “this”, to make them clearer.

Figure 2. The vertical labels are too small to read. Very painful. Make the labeling shorter and 3 x times larger letters.

Include minor tickmarks on a daily basis.

Our response: We have changed the figures appropriately.

No vertical error bars?

Our response: It would be impossible/unnecessary to add error bars to this figure. Errors are discussed in the text.

Line 245. Again, Svalbard glaciers vs Antarctic glaciers? No summer melt in the latter. Please place in proper context.

Our response: We have changed this as suggested

The rest of the discussion is good, albeit a bit long.

Our response: Thank you for these comments, we have taken this on board and removed some parts and added details about testing the model.

Line 429. Reference for first sentence?

Our response: This is now removed as it was confusing

Generally speaking, keep the present tense for describing your results, and the past for already published material, otherwise the reader is confused.

Our response: We have changed as suggested

Line 446. "WE have proposed"  "We propose "

Our response: Changed as suggested

Line 447. I still do not know Glasweb. May be water pressure and air temperature probes?

Our response: They are now referred to as the *in situ* probes.

Now on broader terms, you identify different glacier regimes in response to different subglacier hydrology regimes. That's great because it is practically impossible to sample the subglacial hydrology. To make the point that this matters, I would ask the following: have you documented on how a particular subglacial regime affects projections of mass loss vs another one? In other words, is it well known how the selection of a subglacial regime will affect the projections of sea level rise? More important, can you demonstrate that if you make a mistake, you cannot replicate the glacier evolution, whereas if you make the right choice, it will be easy to reconstruct the glacier evolution? My point is that the impact of the results may be a bit oversold. I agree that this is a very elegant way to categorize the glaciers, based on physical processes. It is less well clear how much this will affect reconstructions and projections. My two cents.

Our response: The modelling suggested is beyond the scope of the paper (and the 5000 word limit). That is the next step in the project. Hopefully we have now made this clearer, and our results are now not oversold.

This paper remains an impressive examples of dense observation network on a set of glaciers, combined with satellite remote sensing.

Our response: We thank the referees for their helpful comments.

Please find enclosed our point-by-point response to the referees' comments. The referees' comments are in italics and our replies are in normal text.

Reviewer comments:

Reviewer #3 (Remarks to the Author):

Thank you for the revision. Generally speaking, it is not helpful for a reviewer to read in the rebuttal letter a statement that says "we made some changes". I do not know what changes are made and where you implemented it. So you are implicitly asking me to go find out. It would be more efficient to indicate what you change, why, and where, e.g. a line number in the MS.

The abstract has a mix of sentences in the present tense (correct) and past (not good to change style). I mentioned that issue before, but you only changed it in a few places. Either you ignore my recommendation, or you implement it in full, including the abstract. I suspect the co-authors of the paper, who include senior authors, could help by reading the rebuttal and revised manuscript. The fact that these changes did not occur suggest that only one author worked on the revision. It would be a good mentoring to help the first author.

Our response: We have now checked the tenses.

Line 27-28. Add freshwater resources. Mountain glaciers matter for sea level, but more so in terms of freshwater resources to local populations in the Andes and Himalayas.

Our response: Now included as suggested line 26.

Line 55. Lower pressure, lower basal friction, more speed and faster retreat. Is that so? When effective pressure is high, ice is compressed against the bed, and friction is low. When the pressure is low, ice is less pressed against the bed and friction is lower. As water pressure increases, the effective pressure decreases, and basal friction is lower. So are you saying "water pressure" or "effective water pressure"? One P_{water} , the other is $P_{ice} - P_{water}$. This is better worded at line 457. Please make sure you say it right because this sentence has major implications for your work.

Our response: We have added the wording "higher water pressure and so lower effective pressure" to clarify this point line 56.

Introduction: You do not want to mention ocean-ice interaction as a key control in addition to subglacial hydrology and likely more important because "it is beyond the scope of your paper"? Well, this is common knowledge. It is not a matter of changing your scope, it is a matter of placing your study into a broader context. If you do not agree, that is one thing. If you say it is beyond the scope, I do not agree with you. Breiðamerkurjökull seems to be a tidewater glacier (line 314, is that correct? The image in figure 1 shows no obvious connection with the ocean) while the others are lake terminating or land terminating, so ocean-ice interaction has an impact on its evolution that you must consider (e.g., migration of the line of grounding, seasonal melt along the ice face depending on subglacial water plume and ocean temperature, tidal-modulation of the horizontal speed, etc.?). If

ending only in a freshwater lake, there are still water-ice interactions that matter. In fact, you say so at line 300.

Our response: We have added this point to the Introduction Line 31-33.

Discussion. Present tense would make it better - again.

Our response: We have checked the tenses and now use the same style in each paragraph.

Results: Describe your new work with the present tense. Published work can be in the past tense. This approach will make your text livelier. Do it in the methods as well. You can choose not to do it, of course, your decision, but you cannot mix styles within the same paragraph.

Our response: We have checked the tenses and now use the same style in each paragraph.

Last comment: "the modeling is beyond the scope of the paper". Well, this is related to the broader impact of your study. You need to discuss how the classification of glacier into different subglacial regimes might affect – or not – glacier modeling and/or projections. I am not asking you to conduct extra work here, e.g., a specific modeling study, but rather to cite literature or past work that would indicate how impactful this will be. If this classification will impact model projections at the 0% or 100% level, it is good to know, on a best effort basis, to understand why you need to distinguish these regimes in the first place. At face value, Breiðamerkurjökull differs from the others due to the tidally connected lake. Perhaps that connectivity matters more than you think. Overall, my burning question remains: why should we care about the type of subglacial hydrology if it does not impact projections?

Our response: We have now added this into the Introduction (Line 58-61) and Conclusion (Line 473-475).

Data availability: Good job with the web site and data access/sharing. Would be nice to get a DOI at the time of submission.

Our response: DOI now requested.

Figures are much improved. I am not 100% sure about having both water pressure and overburden pressure. Perhaps the latter is most relevant.

Our response: We thank the referee for their comments. We have changed the figures to make them clearer, and so have only included overburden pressures in Figures 2 and 3, and water pressure in Figure 4.

Minor ones:

Line 116. What do you mean by averaging velocity over the whole glacier? Do you mean across the glacier width? At what location do you average the satellite-derived velocity w.r. to the ELA? At the ELA (max speed)? At the calving front – when there is max speed if calving into a lake otherwise you get zero. The text says "along the center line", which I find puzzling. I would prefer ELA or calving front.

Our response: We have now added a sentence on the method.

Line 340. Typo 50

Our response: Reference corrected.

Reviewer #4 (Remarks to the Author):

Dear Jane and co-authors,

thanks a lot for this interesting paper. The study compares the glacier velocity behaviour (seasonal and diurnal) of four different glaciers located in Norway and Island. The measurements of glacier velocity (GNSS and from satellite data), meteorological data (air temperature), water pressure, ice thickness, and till grain size were visually compared to categorized the subglacial hydrology into braided, drained braided, and channelized drainage system. The annual process of the four glaciers are very detailed and comprehensive explained and combined with the water drainage and storage. This paper is very valuable for the understanding of subglacial hydrology and provides new insights into the processes regarding hydrology, bed deformation, and glacier velocity. There are rarely as many parameters available as here to visualise the entire process in such detail. I would address some concerns and remarks. It would be great if you could include them in the newer version to make your manuscript better readable and understandable and on the other side to strengthen your statements:

1. The abstract starts with the measurements and not with the background of the problem or why this study is important. Your list all measurements, but it is not clear which parameter are exactly measured. Additionally, the results or the categorization of the drainage is not cleared explained (which types exists and where are the differences) and especially how satellite data could use to recognize the different hydrology types is not well described. Please, revise the abstract including my three above mentioned points. Avoiding repetitions at the beginning of sentences (first three sentences start with 'we') would improve the abstract immense.

Our response: We have changed the abstract as recommended.

2. Table 1 indicated that the parameter like the velocity is available for five years, but the plots present just one year. Is there a reason why you selected just a specific year and showing all the other years? Additionally, some statistics would be essential to strengthen your observations.

Our response: Figure 2 shows measurements from a variety of data sources, some of which suffered data loss, so we show the year with the most complete data set. This information is now included in the text. For Figure 3, we have a full data set, but it would be too much to show every year, so we have chosen one year for comparative purposes, and we have added some statistics to show that the pattern for each year was very similar.

3. In the introduction, you mention the distributed and the channelized drainage. But in Figure 5, just the braided, drained braided, and channelized drainage are displayed and mentioned in the discussion. Is the distributed drainage not relevant in this categorization, is it replaced by the term 'braided', or is it just relevant for hard glacier bed? For me it was little bit confusing and it would be great to explain all these definitions in more detail in the introduction. This would be very helpful for the reader and enhance the comprehension – as well if this study focuses only on soft glacier beds (if yes you could add this in the title) and what the difference between continuum and separated drainage is.

Our response: We have thought about this and changed our language for clarity, we have also changed the title.

Minor corrections:

Line 28: change to "The higher air temperature". If you could change temperature to air temperature, the reader knows that you mean the air temperature and not eventually the glacier temperature. Please take care of it in the complete manuscript.

Our response: Changed throughout the manuscript.

Line 58: Repetitions at the beginning of sentences ('We').

Our response: Rephased.

Line 66-68: Do you have a reference to prove this?

Our response: This is the journal style, the conclusions go at the end of the introduction. More details added about this later in the text.

Line 72: Change to "lower water storage"

Our response: Changed as requested

Line 69-74: Do you have a reference to prove this?

Our response: This is the journal style, the conclusions go at the end of the introduction. More details added about this later in the text.

Line 111: Do you mean 2017/18?

Our response: Yes, now corrected

Line 117: Was the surface velocity derived from the satellite data or using GNSS?

Our response: It was from both, so the text is now corrected.

Line 125-126: But in Figure 2a, water pressure increases as well as velocity during summer? I have the feeling that plot and description does not match or am I wrong?

Our response: We were referring to the summer relative to the other seasons, so have now qualified this.

Line 126: repetition of also (used twice in one sentence).

Our response: Corrected

Line 148: What causes the ice-quakes?

Our response: Detailed discussion of the ice-quakes is beyond the scope of this paper. They are discussed in the reference cited (Hart et al., 2019), but they are from the bed which we have added to the text.

Line 149: How can the GPR recognise that it is a braided subglacial hydrology?

Our response: Details are added to the text.

Line 182-192: Please could you refer the corresponding figures? I assume you refer to Figure 2b and 4b?

Our response: This is now added.

Line 188: Do you mean that the air temperature rose above the spring temperatures? This sentence is not clear for me.

Our response: There was a missing word. It should have said "spring threshold", it has now been corrected.

Line 198: Do you mean Figure 2c?

Our response: Yes, now corrected.

Line 210: What means the numbers of the probe? Are there relevant for the reader? In Line 234 you are using only the number without the 'B'.

Our response: Some detail about the probes added for the reader + Bs added

Line 212: Please could you indicate the date to the DOY like you did in Line 186? That would make it much easier for the reader to sort the events in the time.

Our response: Added as suggested

Line 224: Can you explain the reason for that double event?

Our response: This is now explained in the Discussion – line 367-377.

Line 226: Spring in lowercase.

Our response: Corrected

Line 244: Cannot find Figure 3d.

Our response: Corrected to Figure 3c.

Line 258: You mention the distributed drainage, but it is not displayed in Figure 5.

Our response: Figure 5 is now clearer.

Line 268: Could you give a reason for the velocity decline?

Our response: This is due to the increased melt being absorbed within the hydrological system, as suggested by other workers that are references. We have changed the text so hopefully this is now clearer.

Line 305 and 340: Reference superscript.

Our response: Corrected

Line 369: Could you indicate the grain size in numbers?

Our response: We have added the mean grain size in numbers into Table 1, but we prefer to use the descriptive version in the discussion.

Line 425 – 430: Would make sense to add phase in brackets to make it easier for the reader to combine text and table.

Our response: The phases are now clearer in the Table

Line 438: What are modern glaciers?

Our response: Changed to 'contemporary'

Line 455: An explanation if the temporal resolution of S1 is sufficient to categorize the glacier into the different hydrology systems would be great!

Our response: We now explicitly indicate this in line 398-400

Line 538: Which mode have you used?

Our response: We used IW (Interferometric Wide Swath) model, this is now added to the text.

Line 543: Which steps include the calibration (geometric and/or radiometric)?

Our response: It was radiometrically calibrated, this is now added to the text.

Line 554: Could you add a sentence how you calculated the RMSE, please?

Our response: Information as to how we calculated the RMSE can be found at L558-564, and in the cited articles. However, we appreciate this may not have been as clear as it could have been in the previous draft, and so we have edited this paragraph accordingly."

Figure 2: Colours of air temperature and velocity can hardly be distinguished. Indicate the complete time period with start and end. The plot is slightly overcrowded and it takes time to understand all the information. Perhaps the plots could be slightly improved graphically to make them more readable (same for Figure 3).

Our response: We have now enlarged the diagram to separate the colours.

Figure 3: Colours of air temperature and velocity can hardly be distinguished. I assume that the velocity was derived from satellite data, hence it would make sense to add this information in the caption. Indicate the complete time period with start and end.

Our response: We have now enlarged the diagram to separate the colours and added the time period .Plus added the comment about velocity to the figure caption

Figure 4e: I assume you mean DOY 19-42 as mentioned in the text.

Our response: Corrected

We thank the referees for their helpful comments.